EMBO
Molecular Medicine

# Orkambi® and amplifier co-therapy improves function from a rare *CFTR* mutation in gene-edited cells and patient tissue

Steven V Molinski[1,2] , Saumel Ahmadi[2,3] , Wan Ip[4], Hong Ouyang[4], Adriana Villella[5], John P Miller[5] , Po-Shun Lee[5], Kethika Kulleperuma[1,2], Kai Du[2], Michelle Di Paola[1,2], Paul DW Eckford[2], Onofrio Laselva[2] , Ling Jun Huan[2], Leigh Wellhauser[2] , Ellen Li[2], Peter N Ray[6], Régis Pomès[1,2], Theo J Moraes[4,7] , Tanja Gonska[4,7] , Felix Ratjen[8] & Christine E Bear[1,2,3,*]

## Abstract

The combination therapy of lumacaftor and ivacaftor (Orkambi®) is approved for patients bearing the major cystic fibrosis (CF) mutation: *ΔF508*. It has been predicted that Orkambi® could treat patients with rarer mutations of similar "theratype"; however, a standardized approach confirming efficacy in these cohorts has not been reported. Here, we demonstrate that patients bearing the rare mutation: c.3700 A>G, causing protein misprocessing and altered channel function—similar to ΔF508-CFTR, are unlikely to yield a robust Orkambi® response. While *in silico* and biochemical studies confirmed that this mutation could be corrected and potentiated by lumacaftor and ivacaftor, respectively, this combination led to a minor *in vitro* response in patient-derived tissue. A CRISPR/Cas9-edited bronchial epithelial cell line bearing this mutation enabled studies showing that an "amplifier" compound, effective in increasing the levels of immature CFTR protein, augmented the Orkambi® response. Importantly, this "amplifier" effect was recapitulated in patient-derived nasal cultures—providing the first evidence for its efficacy in augmenting Orkambi® in tissues harboring a rare CF-causing mutation. We propose that this multi-disciplinary approach, including creation of CRISPR/Cas9-edited cells to profile modulators together with validation using primary tissue, will facilitate therapy development for patients with rare CF mutations.

**Keywords** amplifier; c.3700 A>G; CFTR; CRISPR/Cas9; cystic fibrosis
**Subject Categories** Genetics, Gene Therapy & Genetic Disease; Respiratory System

## Introduction

The cystic fibrosis transmembrane conductance regulator (CFTR/ABCC7) is an ATP- and PKA-dependent chloride channel, regulating chloride and bicarbonate ion flux across apical membranes of polarized epithelial cells in certain tissues (e.g., lung, gut, pancreas; Riordan *et al*, 1989; Bear *et al*, 1992; Howell *et al*, 2004; Rowe *et al*, 2005). The tertiary structure of CFTR is arranged into two membrane-spanning domains (MSDs) with six transmembrane helices (TMs) in each MSD, two intracellular nucleotide binding domains (NBDs), and a regulatory (R) domain (Gadsby *et al*, 2006). The TMs of MSD1 and MSD2 form the channel pore, while the NBDs form the catalytic heterodimer required for nucleotide-dependent channel gating; the R-domain regulates the gating of this ion channel (Hwang & Kirk, 2013). Mutations in the *CFTR* gene [MIM: 602421] cause the autosomal-recessive genetic disease cystic fibrosis (CF). *ΔF508* is the major mutation, accounting for ~90% of all cases worldwide; however, there are many additional less frequent CF-causing mutations as documented in the online databases CFTR1 and CFTR2 (see For More Information section; Mickle & Cutting, 2000; Sosnay *et al*, 2013).

Several studies have previously shown that ΔF508 (in NBD1) alters the folding and thermostability of NBD1 when studied in isolation and disrupts the intramolecular assembly of CFTR,

1 Department of Biochemistry, University of Toronto, Toronto, ON, Canada
2 Programme in Molecular Medicine, Research Institute, Hospital for Sick Children, Toronto, ON, Canada
3 Department of Physiology, University of Toronto, Toronto, ON, Canada
4 Programme in Translational Medicine, Research Institute, Hospital for Sick Children, Toronto, ON, Canada
5 Proteostasis Therapeutics, Cambridge, MA, USA
6 Division of Molecular Genetics, Hospital for Sick Children, Toronto, ON, Canada
7 Division of Paediatrics, University of Toronto, Toronto, ON, Canada
8 Division of Respiratory Medicine, Hospital for Sick Children, Toronto, ON, Canada
*Corresponding author. Tel: +1 416 813 5981; E-mail: bear@sickkids.ca

including the structurally relevant interface with the fourth intracellular loop (ICL4) in MSD2, as well as the catalytic NBD1:NBD2 heterodimer (Serohijos *et al*, 2008; He *et al*, 2010; Thibodeau *et al*, 2010; Mendoza *et al*, 2012; Rabeh *et al*, 2012). These conformational defects lead to impaired forward trafficking through the biosynthetic compartments and retention in the endoplasmic reticulum (ER), defects that define the Class 2 CF-causing mutations (Thibodeau *et al*, 2010; Molinski *et al*, 2012; Farinha *et al*, 2013; Eckford *et al*, 2014). The minute fraction of ΔF508-CFTR molecules that manage to reach the cell surface exhibit altered channel activity and reduced cell surface stability at physiological temperature (Du *et al*, 2005; Serohijos *et al*, 2008; He *et al*, 2010, 2013; Okiyoneda *et al*, 2013). Recently, a pharmacological chaperone (VX-809 or lumacaftor) was found to be partially effective in rescuing the functional expression of ΔF508-CFTR to the cell surface in heterologous expression systems (Van Goor *et al*, 2011; Eckford *et al*, 2014). Together with Kalydeco® (VX-770 or ivacaftor), a drug that enhances channel activity, lumacaftor significantly enhanced the functional activity of ΔF508-CFTR in pre-clinical studies of primary bronchial cell cultures and rectal biopsy-derived organoids (this combination therapy has recently been registered as Orkambi®; Van Goor *et al*, 2011; Dekkers *et al*, 2013; Kopeikin *et al*, 2014). Finally, in clinical trials, this combination led to significant improvement in lung function with a 2.6–4.0% increase in forced expiratory volume (percent predicted) in 1-s (FEV1) for *ΔF508* homozygous patients (Boyle *et al*, 2014; Wainwright *et al*, 2015). However, the combination did not provide a significant improvement in FEV1 for compound heterozygous patients with only one allele of *ΔF508*, a result which the authors attribute to the ~50% of available ΔF508-CFTR in patients with this genotype (Boyle *et al*, 2014).

There are multiple CF-causing Class 2 mutations in addition to *ΔF508* thought to cause intrinsic defects in CFTR folding, assembly and trafficking, although these mutations are relatively rare (Welsh & Smith, 1993; Sosnay *et al*, 2013). The prevailing CF drug discovery paradigm suggests that drugs that are effective in targeting ΔF508-CFTR will be effective in rescuing other Class 2 CFTR mutants (Awatade *et al*, 2015; Rapino *et al*, 2015). However, in testing this hypothesis for a rare CF-causing mutation: c.3700 A>G (p.Ile1234_Arg1239del-CFTR or ΔI1234_R1239-CFTR; Molinski *et al*, 2014), we found that in primary nasal epithelial cultures, the combination therapy of lumacaftor and ivacaftor failed to rescue the mutant's functional expression to a degree considered to be therapeutically relevant.

Here, we show that a companion therapy, an "amplifier" compound that stabilizes *CFTR* mRNA, was effective in augmenting Orkambi® functional enhancement in a CRISPR/Cas9-edited bronchial cell line bearing this rare mutation. Further, we show that these results were recapitulated in patient-derived nasal epithelial cultures. Hence, this work highlights a novel strategy to identify compounds that have the potential to improve the function of CFTR chloride conduction in individuals affected by rare CF-causing mutations. Further, these studies provide the first evidence that an amplifier compound augments the functional enhancement conferred by Orkambi® for a mutation other than *ΔF508*.

# Results

## Molecular dynamic simulations predict consequences of the rare mutation: ΔI1234_R1239, on full-length protein structure and function

We developed a molecular model of the full-length ΔI1234_R1239-CFTR mutant protein in the open-channel state to gain greater insight into the structural and functional defects we observed in biochemical studies (Molinski *et al*, 2014). We conducted all-atom molecular dynamics (MD) simulations, adopting a ground-up approach, using homology models of the full-length wild-type (WT)-CFTR protein and generated ΔI1234_R1239-CFTR from the WT-CFTR template based on the Sav1866 crystal structure (Fig EV1A, left panel; Dawson & Locher, 2006, 2007). Multiple simulations with multiple repeats were performed on both WT-CFTR and ΔI1234_R1239-CFTR, starting from the equilibrated WT-CFTR system; total simulation time for each system was over 0.5 μs. The residues deleted in the mutant are predicted to constitute a loop connecting two anti-parallel β-strands (β2 and β3) in NBD2, which form part of an anti-parallel three-stranded β-sheet and a parallel four-stranded β-sheet, respectively (Fig 1A). Importantly, this secondary structure prediction was recently validated by the cryo-electron microscopy-derived structure of human CFTR (Fig EV1A, right panel; Liu *et al*, 2017). After 30 ns of unrestrained simulation, the loss of secondary structure in β-strands β2 and β3 with varying intensities was evident in multiple simulations of ΔI1234_R1239-CFTR, while both of these strands remained intact in simulations of WT-CFTR. Further, one or both of the two β-strands (β2 and β3) unraveled in 2/3 of the simulations of ΔI1234_R1239-CFTR, consistent with the deletion significantly impacting the secondary structure that flanks it (Fig 1A). We then analyzed how these local perturbations may affect the overall conformation of NBD2. The root-mean-square deviation (RMSD) of the backbone of each NBD2 residue from the starting conformation of NBD2 as a function of its residue number was compared between WT-CFTR and ΔI1234_R1239-CFTR systems (Fig 1B). The deviation of all regions in ΔI1234_R1239-CFTR falls within the error of the WT-CFTR system. These findings suggest that in this model of the full-length protein, the effect of ΔI1234_R1239 was localized to the immediate vicinity of the deletion and did not modify the global fold of the whole NBD2 during the relatively short simulation time of 30 ns.

To examine the effect of ΔI1234_R1239 on the canonical ATP binding site of NBD2, which contains the Walker A lysine, K1250, we compared the conformational ensemble of the replicas of the WT-CFTR and mutant protein, using the ATP-analog-bound crystal structure of Sav1866 as a reference (Fig EV2; Dawson & Locher, 2006, 2007; Dalton *et al*, 2012). The superimposition of backbone atoms of Walker A in WT-CFTR and ΔI1234_R1239-CFTR on the crystal structure revealed more structural heterogeneity in the mutant protein than in WT-CFTR. Moreover, the spatial distribution of the ammonium N atom of K1250 of the Walker A motif in NBD2, which is critical for ATP binding, shows a wider dispersion within the mutant ensemble compared to WT-CFTR, for which more replicas were comparable to the reference. These findings predict that the canonical ATP binding site of NBD2 that engages in ATP-dependent heterodimerization with NBD1 and ATP-dependent channel gating is structurally perturbed in ΔI1234_R1239-CFTR. Such predictions are consistent with the defects in protein processing (see

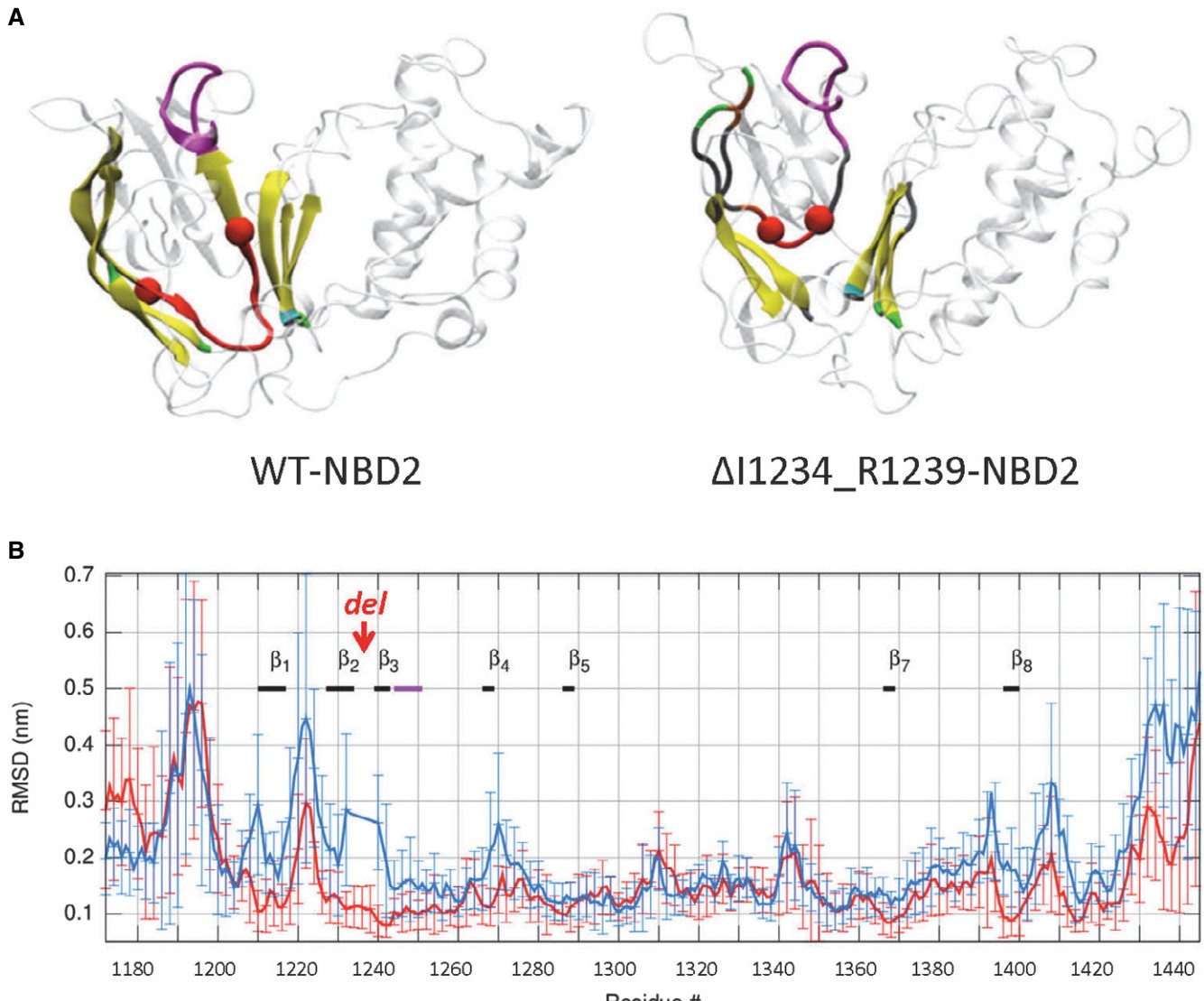

**Figure 1.  c.3700 A>G mutation (ΔI1234_R1239) is predicted to induce local conformational defect in NBD2 of CFTR in molecular dynamic simulations.**

A   Representative snapshots of NBD2 in WT-CFTR (left panel) and ΔI1234_R1239-CFTR (right panel) systems following 30 ns of simulation. Each secondary structure element of the labeled β-strands is represented by a unique color as follows: isolated-bridge, brown; coil, gray; β-strand, yellow; 3–10 helix, dark blue; α-helix, cyan; turn, green; Walker A is in magenta, while the rest of NBD2 is in transparent representation. The amide N and carbonyl C atoms of residues 1,233 and 1,240, respectively, are shown as red spheres, and the backbone atoms between the same residue pair are shown in red cartoon representation.

B   Replica-averaged root-mean-square deviation (RMSD) of backbone atoms of NBD2 from the starting conformation of WT-CFTR and ΔI1234_R1239-CFTR as a function of the residue number is represented in red and blue thick lines, respectively. The six-residue deletion (ΔI1234_R1239) is labeled "del". The thick black and magenta lines denote β-strands and the Walker A motif, respectively.

Fig EV1B and C) and channel function previously reported for this mutation.

## Misprocessing and altered function of ΔI1234_R1239-CFTR can be partially ameliorated by small molecule modulators of ΔF508-CFTR

As previously published, after expression in human embryonic kidney (HEK)-293 cells, the ΔI1234_R1239-CFTR mutant is retained in the ER (immature form, called band B) and lacks the complex-glycosylated form observed for the WT-CFTR protein (i.e., mature, band C form; Fig EV1B; Molinski *et al*, 2014). Similar to ΔF508-CFTR, the ER-retained ΔI1234_R1239-CFTR mutant protein undergoes proteasomal degradation (Fig EV3). Interestingly, in contrast to ΔF508-CFTR (Denning *et al*, 1992), low temperature (27°C) incubation and chemical chaperones such a sodium butyrate, while effective in enhancing the total abundance of the mutant protein, did not enhance the maturation of ΔI1234_R1239-CFTR, suggesting that

there may be distinct defects in biosynthesis of this mutant relative to ΔF508-CFTR (Fig EV3; Cheng *et al*, 1995; Moyer *et al*, 1999).

In Fig 2, we show that corrector compounds partially effective in promoting native folding and assembly of ΔF508-CFTR (Okiyoneda *et al*, 2013) are also partially effective in overcoming the misprocessing defect of ΔI1234_R1239-CFTR (Fig 2A and B). Lukacs and collegues categorized ΔF508-CFTR corrector compounds based on their putative mechanism of action with Class 1 compounds acting to repair aberrant intramolecular interactions between ICL4 and NBD1 and stabilize MSD1 (Loo *et al*, 2013; Ren *et al*, 2013; Laselva *et al*, 2016), Class 2 compounds enhancing stability of NBD2, and Class 3 compounds promoting stability of NBD1 (He *et al*, 2013; Okiyoneda *et al*, 2013). For ΔF508-CFTR, the combination of correctors from each class led to full correction of its protein processing defect and the appearance of wild-type levels of mature protein (Okiyoneda *et al*, 2013). In the case of ΔI1234_R1239-CFTR, Class 1 corrector compounds (VX-661, C18, VRT-325, and VX-809) and the Class 2 corrector: Corr-4a (C4) were effective in enhancing the maturation and functional activity of ΔI1234_R1239-CFTR (Fig 2A–C).

Channel activity of ΔI1234_R1239-CFTR in HEK-293 cells was measured using a halide flux assay (Mansoura *et al*, 1999). As expected, at physiological temperature (37°C), the low percentage of mature, plasma membrane-localized ΔI1234_R1239-CFTR protein was insufficient to mediate significant chloride flux after activation by CFTR agonists, forskolin, and isobutylmethylxanthine (IBMX), findings in agreement with the recent work by Ramalho and collegues (Fig 2C; Ramalho *et al*, 2015). Long-term (24-h) treatment with the Class 1 corrector VX-809 or the Class 2 corrector C4, albeit to a lesser extent, conferred the appearance of cAMP-activated chloride flux (Fig 2C). Interestingly, the magnitude of the VX-809-rescued channel function for ΔI1234_R1239-CFTR was approximately half of that observed for VX-809-rescued ΔF508-CFTR despite similar rescue of the processing defect (Fig 2B and C). The relatively severe channel function defect for ΔI1234_R1239-CFTR is consistent with the *in silico* studies that predicted a deleterious effect of this deletion on the ATP binding site conferred by NBD2.

These rescue effects suggest that, as for ΔF508-CFTR, compounds with diverse chemical structure and that target different defective steps in mutant CFTR folding have the potential to correct the misprocessing defects induced by ΔI1234_R1239-CFTR. Interestingly, unlike the findings for ΔF508-CFTR (Okiyoneda *et al*, 2013), co-treatment with Class 1 and Class 2 correctors (i.e., VX-809 and C4) did not lead to an additive or synergistic response in processing or functional activity of the ΔI1234_R1239-CFTR mutant. We could not assess the effect of the Class 3 chemical chaperone: glycerol, as it exerted a deleterious effect on total protein expression in the current studies. The lack of synergy between corrector classes suggests that the intrinsic defects in protein folding, assembly, and function caused by ΔI1234_R1239 are not identical to those caused by ΔF508 in CFTR.

### ΔI1234_R1239 induces conformational defects in the full-length protein as predicted by *in silico* studies

To better define the conformational defects in ΔI1234_R1239-CFTR, we conducted limited proteolysis studies of the mutant CFTR expressed in HEK-293 cells. First, we validated our methods by confirming previous studies showing that the full-length as well as NBD2 (a 36-kDa fragment reactive to the NBD2-specific antibody M3A7) of ΔF508-CFTR protein exhibited enhanced protease (trypsin) susceptibility relative to WT-CFTR (Fig 2D; Du *et al*, 2005; Cui *et al*, 2007; Yu *et al*, 2011; Eckford *et al*, 2014). Applying these methods to the full-length ΔI1234_R1239-CFTR protein, we found that this mutant protein also exhibited enhanced protease sensitivity, supporting the idea that its conformation is altered (Fig 2D and E). Interestingly, in the context of the full-length protein, the domain in which the mutation is located (NBD2) is relatively stable and trypsin-resistant even at high trypsin concentrations (Fig 2D). In contrast, the fragmentation pattern for the amino-terminal half of the ΔI1234_R1239-CFTR mutant is strikingly different from that of WT-CFTR and ΔF508-CFTR (Figs 2D and EV4). Among other differences, a dominant 36-kDa fragment corresponding to NBD1 that

---

**Figure 2.   Small molecule correctors of ΔF508-CFTR partially rescue processing and conformational defect of ΔI1234_R1239-CFTR in the HEK-293 expression system.**

A   Immunoblots of steady-state expression of ΔI1234_R1239-CFTR following treatments with ΔF508-CFTR modulators. Class 1 and Class 2 correctors are labeled in red and blue, respectively. ΔF508-CFTR and WT-CFTR are included as controls.

B   Quantitation of maturation by most efficacious modulators; VX-809 was used as a representative Class 1 corrector. Maturation was benchmarked to ΔF508-CFTR and WT-CFTR (mean ± SEM, $n = 3$ biological replicates, and statistical significance tested using two-way ANOVA with Tukey's multiple comparisons test). For ΔF508-CFTR: DMSO versus VX-809, **$P = 0.0046$. For ΔI1234_R1239-CFTR: DMSO versus VX-809, **$P = 0.0032$; DMSO versus C4, ***$P = 0.0009$; DMSO versus VX-809 + C4, ***$P = 0.0005$. For ΔF508-CFTR (DMSO) versus ΔI1234_R1239-CFTR (DMSO), **$P = 0.0022$.

C   Quantitation of rate of activation (chloride efflux assay) of ΔI1234_R1239-CFTR following chronic treatment with ΔF508-CFTR correctors and acute activation (forskolin/IBMX). Activation rate was benchmarked to ΔF508-CFTR and WT-CFTR (mean ± SEM, $n = 4$ biological replicates, and statistical significance tested using two-way ANOVA with Tukey's multiple comparisons test). For ΔF508-CFTR: DMSO versus VX-809, ***$P = 0.0003$. For ΔI1234_R1239-CFTR: DMSO versus VX-809, **$P = 0.0018$; DMSO versus C4, *$P = 0.0408$; DMSO versus VX-809 + C4, *$P = 0.0199$. For ΔF508-CFTR (VX-809) versus ΔI1234_R1239-CFTR (VX-809), *$P = 0.0222$.

D   Immunoblots of proteolytic digestion of full-length WT-CFTR, ΔF508-CFTR and ΔI1234_R1239-CFTR. Band B, black arrowhead; band C, white arrowhead. Square brackets denote NBD1 and NBD2 fragments of interest, and those generated from ΔI1234_R1239-CFTR were quantitatively analyzed in panel (G).

E   Quantitation of proteolytic susceptibility of full-length WT-CFTR, ΔF508-CFTR and ΔI1234_R1239-CFTR (mean ± SEM, $n = 3$ biological replicates, and statistical significance tested using unpaired *t*-tests). For 1.6 μg/ml trypsin: ΔF508-CFTR versus ΔI1234_R1239-CFTR, *$P = 0.0206$.

F   Quantitation of proteolytic susceptibility of ΔI1234_R1239-CFTR in the absence (DMSO) or presence of small molecule correctors of ΔF508-CFTR (i.e., VX-809, C4, or VX-809 + C4; mean ± SEM, $n = 3$ biological replicates, and statistical significance tested using unpaired *t*-tests). For ΔI1234_R1239-CFTR (3.1 μg/ml trypsin): DMSO versus VX-809, *$P = 0.0151$; DMSO versus C4, *$P = 0.0458$; DMSO versus VX-809 + C4, *$P = 0.0422$. For ΔI1234_R1239-CFTR (6.3 μg/ml trypsin): DMSO versus VX-809, *$P = 0.0412$; DMSO versus C4, *$P = 0.0233$; DMSO versus VX-809 + C4, *$P = 0.0308$.

G   Quantitation of proteolytic digestion of NBD1 and NBD2 fragments of ΔI1234_R1239-CFTR in the presence of small molecule correctors (VX-809, C4, or VX-809 + C4; mean ± SEM, $n = 3$ biological replicates, and statistical significance tested using two-way ANOVA with Tukey's multiple comparisons test). For NBD1: DMSO versus C4, *$P = 0.0141$; DMSO versus VX-809 + C4, *$P = 0.0200$; C4 versus VX-809 + C4, *$P = 0.0353$.

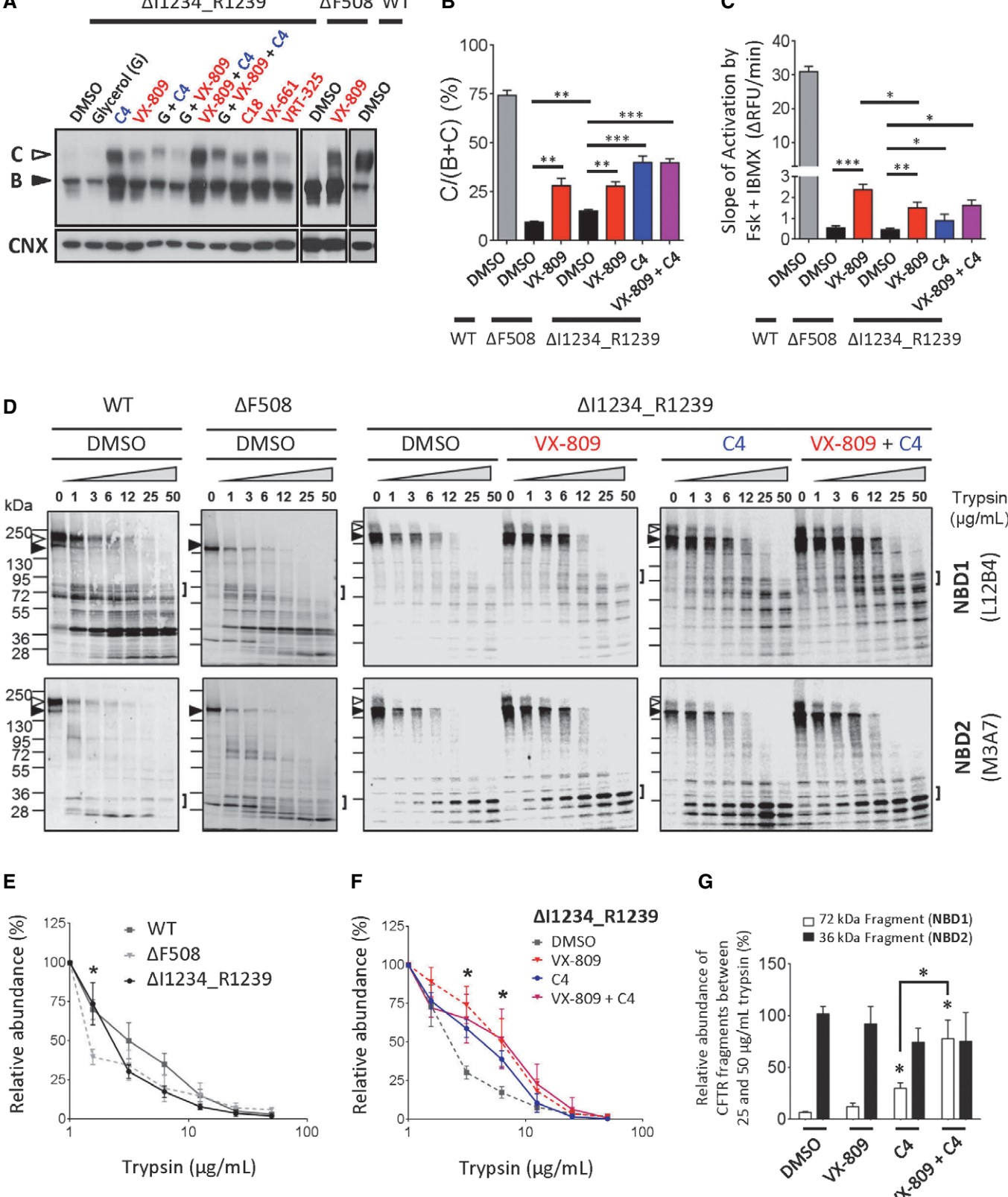

**Figure 2.**

is trypsin-resistant in WT-CFTR or ΔF508-CFTR is undetectable for ΔI1234_R1239-CFTR. Although this approach does not provide molecular detail, the results suggest that in the context of the full-length CFTR protein, the ΔI1234_R1239 mutation does not significantly impact the overall stability of NBD2, the domain that harbors the mutation. Rather it confers an allosteric, de-stabilizing effect on the contralateral half of CFTR containing NBD1. Importantly, these results are consistent with the predictions generated in molecular dynamic simulation studies that the heterodimerization of NBD1 with NBD2 will be disrupted by ΔI1234_R1239-CFTR in the context of the full-length protein.

### Modulators of ΔF508-CFTR are partially effective in rescuing the conformational and functional defects of ΔI1234_R1239-CFTR

Analyses of protease susceptibility studies (Fig 2D, F and G) suggest that the corrector compounds VX-809 or C4 improved ΔI1234_R1239-CFTR processing (Fig 2A) by increasing its assembly and compactness. As shown in Fig 2F, the relative abundance of full-length mutant protein in the presence of low trypsin concentrations was significantly greater in cells that had been pretreated with VX-809, C4, or VX-809 + C4. This effect was mediated by directly or indirectly stabilizing the amino-terminal half (the 72-kDa fragment) of the mutant protein (Fig 2G).

We then assessed the response of ΔI1234_R1239-CFTR channels having undergone correction using VX-809 for their response to the channel potentiator VX-770, as this potentiator can act independently of the Walker A motif in NBD2 (Yeh *et al*, 2015). As in the case of ΔF508-CFTR, VX-809-corrected ΔI1234_R1239-CFTR channels were potentiated by acute treatment with VX-770 (1 μM) after pre-treatment with forskolin (Fig 3A and B). However, paired experiments showed that this potentiation was significantly reduced to ~50% of that observed in cells expressing ΔF508-CFTR (Fig 3A and B).

### Chronic treatment with VX-770 exerts deleterious effects on the modest correction of ΔI1234_R1239-CFTR by VX-809

In light of reports demonstrating that long-term (24–48 h) treatment with VX-770 *in vitro* diminishes the pharmacological correction of ΔF508-CFTR (Cholon *et al*, 2014; Veit *et al*, 2014), we investigated the consequences of long-term VX-770 treatment on the functional correction of ΔI1234_R1239-CFTR-mediated by VX-809. Similar to the findings with ΔF508-CFTR, we found that a 24-h treatment with VX-770 (10 μM) significantly reduced correction of the processing defect in ΔI1234_R1239-CFTR by VX-809 (Fig 4A and B) and exerted a significant deleterious effect on its functional activity in this overexpression system (Fig 4C and D). Hence, ΔI1234_R1239-CFTR (and possibly other Class 2 mutant CFTR proteins) may be similar to ΔF508-CFTR with respect to the detrimental effect of chronic *in vitro* dosing of high concentrations of VX-770 (10 μM) on its stability.

### ΔF508-CFTR correctors induce modest rescue in primary nasal epithelial cells from individuals homozygous for c.3700 A>G (ΔI1234_R1239) relative to nasal cultures from individuals who are homozygous for ΔF508

Our next goal was to determine whether the functional rescue observed in our heterologous expression system translated to a detectable response in patient tissue. Therefore, we generated differentiated nasal epithelial cultures from two CF individuals (siblings homozygous for ΔI1234_R1239) as well as cultures from four other non-CF family members (homozygous for WT-CFTR or carriers of ΔI1234_R1239). We excluded two individuals who are heavy smokers as cigarette smoke is known to be deleterious to CFTR expression (Cantin *et al*, 2006). Immunofluorescence studies of cultures generated from a non-CF family member show well-polarized epithelium with apically localized WT-CFTR, while

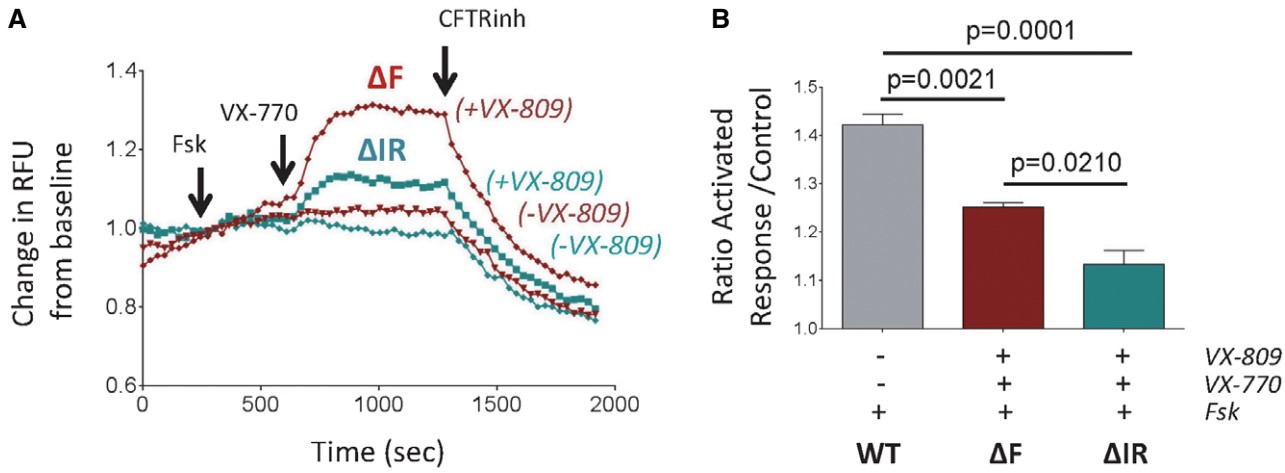

**Figure 3.  VX-809 and VX-770 partially rescue functional expression of ΔI1234_R1239-CFTR in a HEK-293 expression system.**

A   Representative traces (membrane depolarization assay) of ΔI1234_R1239-CFTR (ΔIR) and ΔF508-CFTR (ΔF) function following chronic treatment with VX-809 and acute activation (forskolin/VX-770).

B   Quantitation of relative activated responses of ΔI1234_R1239-CFTR and ΔF508-CFTR (mean ± SEM, *n* = 3 biological replicates). Activation of WT-CFTR by forskolin (without added compound, *n* = 3, shown for comparison). Statistical significance tested using two-way ANOVA with Tukey's multiple comparisons test.

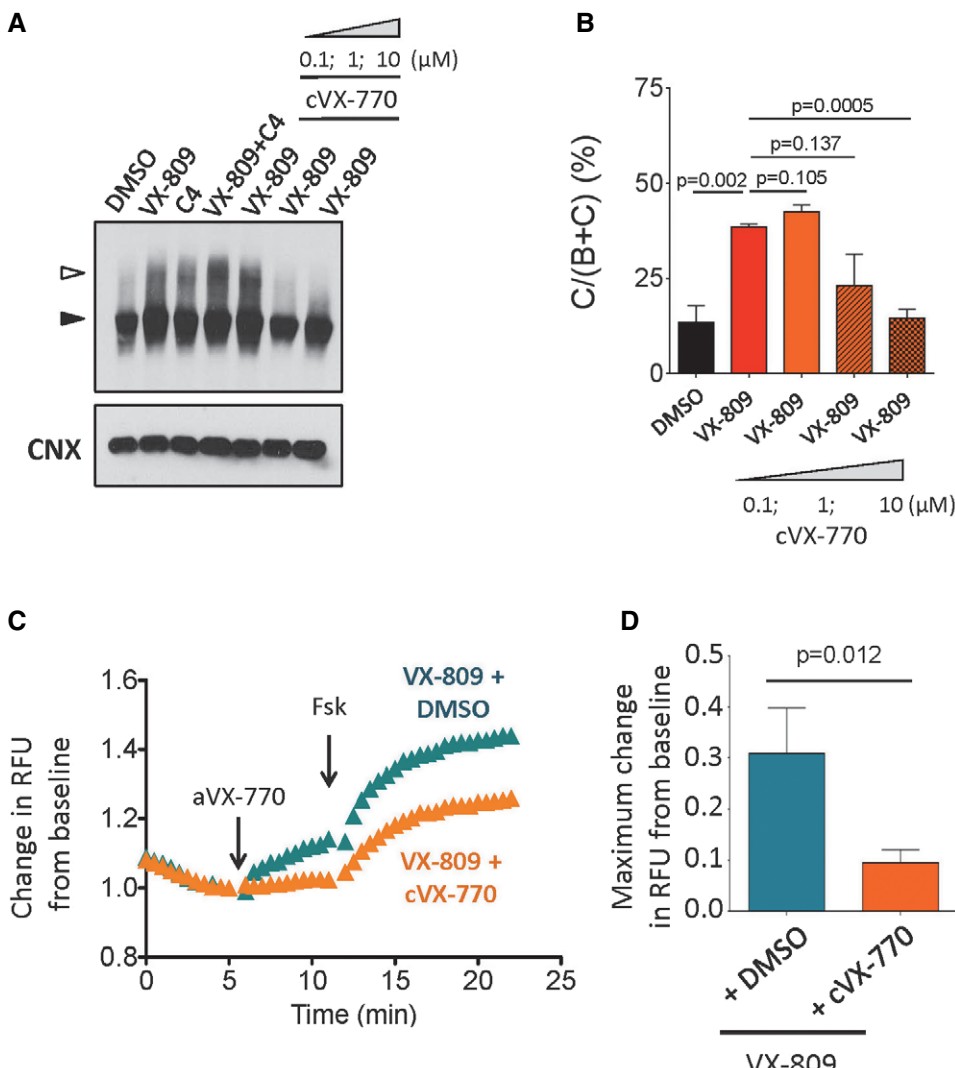

**Figure 4. Potential deleterious effect of VX-770 at high concentrations observed for ΔI1234_R1239-CFTR in a heterologous expression system.**

A, B    Immunoblot (A) and quantitation (B) of ΔI1234_R1239-CFTR expression following pharmacological correction (VX-809) in the absence (DMSO) and presence of chronic (24 h) VX-770 treatment (0.1, 1, and 10 μM; mean ± SEM, n = 3 biological replicates). Band B, black arrowhead; band C, white arrowhead. Statistical significance tested using two-way ANOVA with Tukey's multiple comparisons test.

C, D    Representative traces (membrane depolarization assay) (C) and quantitation (D) of ΔI1234_R1239-CFTR function following pharmacological correction (VX-809) in the absence (DMSO) and presence of chronic (24 h) VX-770 treatment (10 μM) and acute activation (VX-770/forskolin; mean ± SEM, n = 8 biological replicates). Statistical significance tested using two-way ANOVA with Tukey's multiple comparisons test.

cultures from CF-1 show that ΔI1234_R1239-CFTR protein was absent from apical membranes (Fig 5A).

Immunoblots showed that in native tissues, mutant CFTR protein was misprocessed as expected (predominantly immature, band B) and total abundance (bands B and C) was drastically reduced relative to total abundance in cultures from a non-CF family member (heterozygous for the mutation; Fig 5A and B.i and B.ii). On the other hand, abundance of an alternative, apical chloride channel, TMEM16A, is similar for all cultures, arguing that the previously described differences in CFTR expression among nasal cultures are dictated by *CFTR* genotype rather than difference in culture quality (Fig 5Bi). As shown in Fig 5A and B, VX-809 or VX-661 treatment increased the abundance of mature WT-CFTR in

non-CF nasal cultures by approximately twofold (Fig 5Biii). Similarly, the abundance of the immature (band B) form of ΔI1234_R1239-CFTR in nasal cultures from CF-1 increased by approximately twofold (Fig 5Biv). However, rescue of the mature form (band C) was not consistently detected in CF-1 cultures compared to non-CF cultures.

To evaluate the functional competency of ΔI1234_R1239-CFTR in primary tissues, electrophysiological studies were performed on nasal epithelial cultures from CF-1 and CF-2 and subsequently compared to those from non-CF family members (Fig 6A). As expected, forskolin treatment led to a robust change in the equivalent chloride current (Ieq) in monolayers of nasal cells from a non-CF individual, and this forskolin-induced current was sensitive to

inhibition by CFTRinh-172, confirming that it is mediated by the CFTR channel (Fig 6B). Interestingly, one non-CF family member failed to exhibit a forskolin response but did exhibit a robust CFTRinh-172 response (22.45 $\mu$A/cm$^2$), suggesting that CFTR was basally active in this nasal epithelial culture (Fig 6C). We next determined that both CF affected individuals, CF-1 and CF-2, exhibited abrogated CFTR channel function relative to those mediated by non-CF family members (Fig 6C).

A 48-h treatment with lumacaftor (VX-809) induced a modest but significant increase ($P = 0.01$, $n = 6$) in VX-770 potentiated CFTR activity in the nasal cultures of CF-1 (Fig 6D and E). A similar treatment failed to increase CFTR channel activity in cultures derived from CF-2 ($P = 0.06$), although there was a positive trend. Importantly, the VX-809 corrected and VX-770 potentiated channel activity measured in six cultures from patient CF-1 is significantly less ($P = 0.01$) than the activity observed in nasal cultures obtained from six patients who are homozygous for the $\Delta$F508 mutation and similarly treated (Fig 6E). Together, these *in vitro* findings suggest that the combination of VX-809 and VX-770 may not be as effective in individuals with this "protein processing" mutant than it is for patients harboring $\Delta$F508.

### An alternative chloride channel is functionally expressed in nasal cultures generated from the affected family members

As previously mentioned, the calcium-activated chloride channel (CaCC), TMEM16A, is also expressed in the primary nasal cultures as determined by immunoblotting (Fig 5B; Caputo *et al*, 2008; Schroeder *et al*, 2008; Huang *et al*, 2009). Given the poor response of $\Delta$F508-CFTR modulators on $\Delta$I1234_R1239-CFTR, we were prompted to determine whether a putative, alternative therapeutic target, that is, CaCC, was functionally expressed in nasal cultures derived from the two affected individuals (Caputo *et al*, 2008; Kunzelmann *et al*, 2012). We found that nasal cell cultures from CF-2 exhibited robust responses to ATP, an agonist of the P2Y2 receptor (Fig 6F; Rajagopal *et al*, 2011). These studies suggest that the nasal cultures generated from both CF patients were capable of exhibiting epithelial chloride secretion, supporting their relevance in drug testing and providing rationale for developing agonists for TMEM16A as a potential therapy for these affected CF patients (Namkung *et al*, 2011).

### CRISPR/Cas9-edited HBE cell line recapitulates endogenous expression of $\Delta$I1234_R1239-CFTR and response to $\Delta$F508-CFTR modulators

The accessibility of primary nasal cultures is relatively limited, although new procedures are being tested that would enhance their potential for expansion and capacity for continued passaging. Hence, there is a need to establish a renewable cellular model which, unlike the HEK-293 expression system, recapitulates the basic defects of reduced total CFTR protein expression and processing defined in native tissues. Accordingly, we edited human bronchial epithelial (HBE) cells endogenously expressing WT-CFTR to bear the $\Delta$I1234_R1239 mutation on both *CFTR* alleles using CRISPR/Cas9 (Fig EV5). To assess the fidelity of this new model (HBE-$\Delta$IR) in recapitulating the protein defects caused by $\Delta$I1234_R1239 that was observed in patient-derived nasal epithelium, we compared the steady-state levels of the mutant protein in the edited cell line to

WT-CFTR expressed in the parental HBE cell line (Fig 7A). The $\Delta$I1234_R1239 mutation led to a profound reduction in the total amount of CFTR protein, defective processing, and a low ratio of mature/immature protein (i.e., band C/B + C), all comparable biochemical phenotypes to those observed in primary tissues (Fig 7A, left panel). We also found that relative levels of total $\Delta$I1234_R1239-CFTR protein were comparable those expressed in patient CF-1 (Fig 7A, right panel). We next tested the functional activity of $\Delta$I1234_R1239-CFTR in the edited cell line. As expected, relative levels of activity for this mutant were less than WT-CFTR, but greater than a CFTR knockout (KO) cell model, similarly created using the CRISPR/Cas9 technology (Fig 7B). Further, as shown in analyses of the primary tissues for affected individuals (CF-1 and CF-2) treated with VX-809, we detected only a modest increase in total protein abundance with treatments of the HBE-$\Delta$IR cell line (Fig 7D). We interpret these findings to suggest that the HBE-$\Delta$IR cell line offers a robust model with which to test additional interventions aimed at improving its functional expression in relevant tissues.

### A small molecule amplifier (PTI-CH) augments the effect of the corrector (VX-809) and potentiator (VX-770) on $\Delta$I1234_R1239-CFTR in CRISPR/Cas9-engineered and patient-specific tissues

As previously shown (Fig EV3), the HDAC inhibitor, sodium butyrate, was effective in enhancing the immature form (band B) of the $\Delta$I1234_R1239-CFTR protein expression in HEK-293 cells and we confirmed that the related compound, 4-phenylbutyrate (4-PBA), was also effective in increasing the total steady-state level of band B of the mutant protein in the edited bronchial epithelial cell line: HBE-$\Delta$IR (Fig EV6; Rubenstein *et al*, 1997). This positive effect of 4-PBA primarily reflects an increase in mutant CFTR transcript abundance (Fig EV6). Further, co-administration of 4-PBA and VX-809 in this cell line led to the appearance of the mature form of the mutant protein with significantly enhanced stability as monitored by cycloheximide chase (Fig EV6). These findings suggest that effective rescue of the mutant protein could be achieved by enhancing its total expression in addition to modulating its defective maturation using VX-809.

We were then prompted to test a novel class of small molecule modulators of CFTR levels called "amplifiers". Amplifiers have been reported to increase *CFTR* mRNA by enhancing its stability and, as a consequence, increase CFTR protein abundance in a mutation-agnostic fashion in pre-clinical studies (Miller *et al*, 2016). To determine whether the amplifier, PTI-CH, from this novel class of compounds increased the stability of *$\Delta$I1234_R1239-CFTR* mRNA, we measured the levels of *CFTR* mRNA in response to PTI-CH in the absence or presence of the transcriptional inhibitor $\alpha$-amanitin in the CRISPR/Cas9-edited HBE cell line. As shown in Fig 7C, PTI-CH treatment increased *$\Delta$I1234_R1239-CFTR* mRNA, and consistent with a post-transcriptional mechanism, the effect size was indistinguishable between treatments in the absence and presence of transcriptional block. This modest increase in mRNA abundance following PTI-CH treatment did not translate to a significant increase in mutant protein on its own, but when PTI-CH treatment was combined with VX-809, a significant increase in both forms (the immature band B and mature band C) was observed (Fig 7Di and Dii). Further, the combination of PTI-CH and VX-809 significantly enhanced the peak mutant

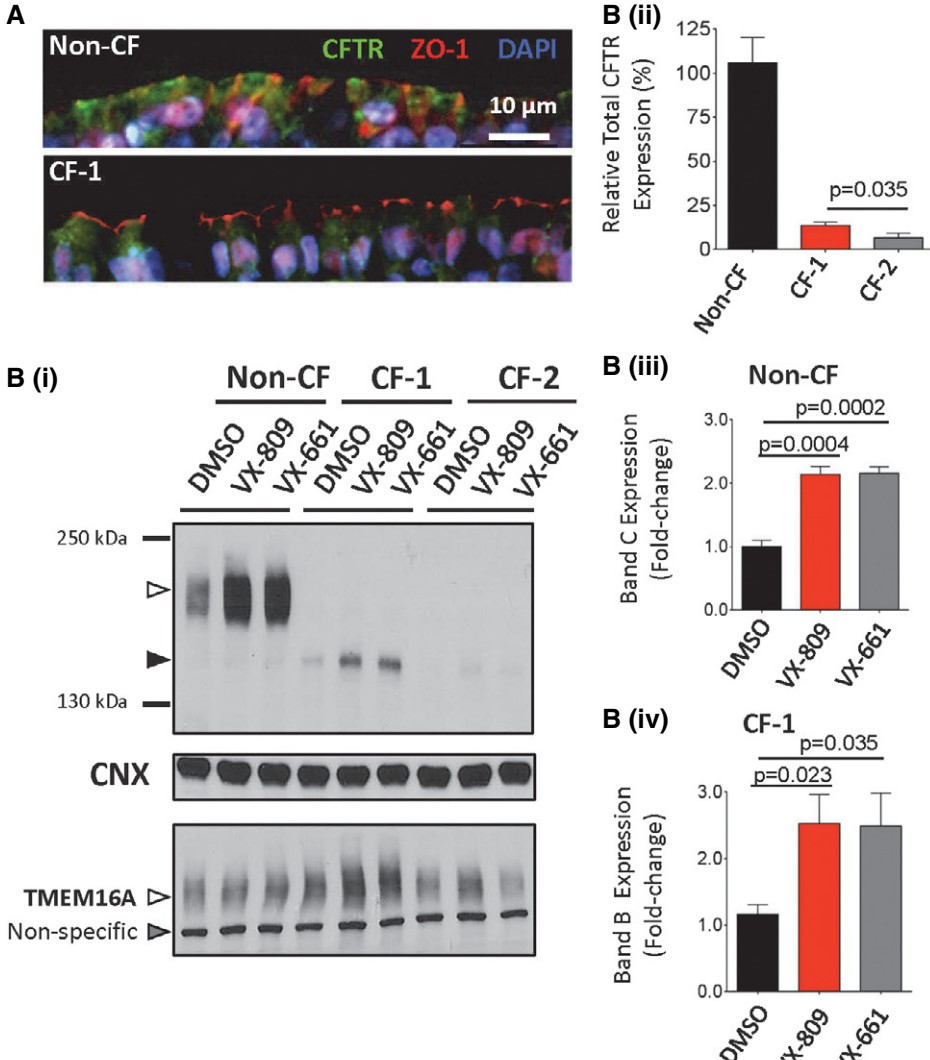

**Figure 5.   Nasal epithelial cultures derived from siblings homozygous for c.3700 A>G (ΔI1234_R1239) exhibit reduced CFTR protein expression relative to non-CF family members with modest rescue by VX-809 or VX-661.**

A     Top panel: Immunofluorescence showing expression and localization of WT-CFTR (green) on primary nasal tissue obtained from a non-CF family member. Tight junction protein ZO-1 (red) and nuclei (DAPI) are also labeled; scale bar represents 10 μm. Bottom panel: Expression and localization of ΔI1234_R1239-CFTR (CF-1, green).

B (i)   Immunoblot of steady-state expression of CFTR (upper blot: band B, black arrowhead; band C, white arrowhead), CNX (middle blot), and TMEM16A (lower blot: non-specific band, gray arrowhead; TMEM16A-specific band, white arrowhead) from non-CF (i.e., WT-CFTR) and CF (i.e., CF-1 and CF-2, homozygous for ΔI1234_R1239-CFTR) following treatment with VX-809 or VX-661.

B (ii)  Quantitation of total CFTR (band C and band B forms in non-CF and CF individuals, respectively) expressed in the absence of small molecules (mean ± SEM, $n = 4$). Statistical significance assessed using unpaired $t$-test.

B (iii) Quantitation of the increase in band C (mature form) of WT-CFTR by VX-809 or VX-661 in cultures from the non-CF individual (mean ± SEM, $n = 4$). Statistical significance assessed using paired $t$-tests.

B (iv)  Quantitation of the increase in band B (immature form) of ΔI1234_R1239-CFTR by VX-809 or VX-661 in cultures from CF-1 (mean ± SEM, $n = 4$). Statistical significance of comparisons assessed using paired $t$-tests.

CFTR channel activity above that achieved with VX-809 treatment alone (Fig 7E and F).

Based on the above findings, we next evaluated the correction effect of VX-809 in combination with the amplifier compound on patient-derived nasal cultures. As predicted on the basis of our studies in the CRISPR/Cas9-edited HBE cell line, treatment of nasal cultures from CF-1 with VX-809 plus PTI-CH (1 μM) led to an increase in the

CFTR-mediated channel function stimulated by forskolin and potentiated by VX-770 (Fig 8A). This increase in activity was associated with increased appearance of band B and band C as studied by Western blotting (Fig 8B). The positive effect of PTI-CH in augmenting the functional improvement mediated by VX-809 in Ussing chamber studies was reproducible and statistically significant for multiple nasal cultures ($n = 6$), derived from two different nasal scrapings from this

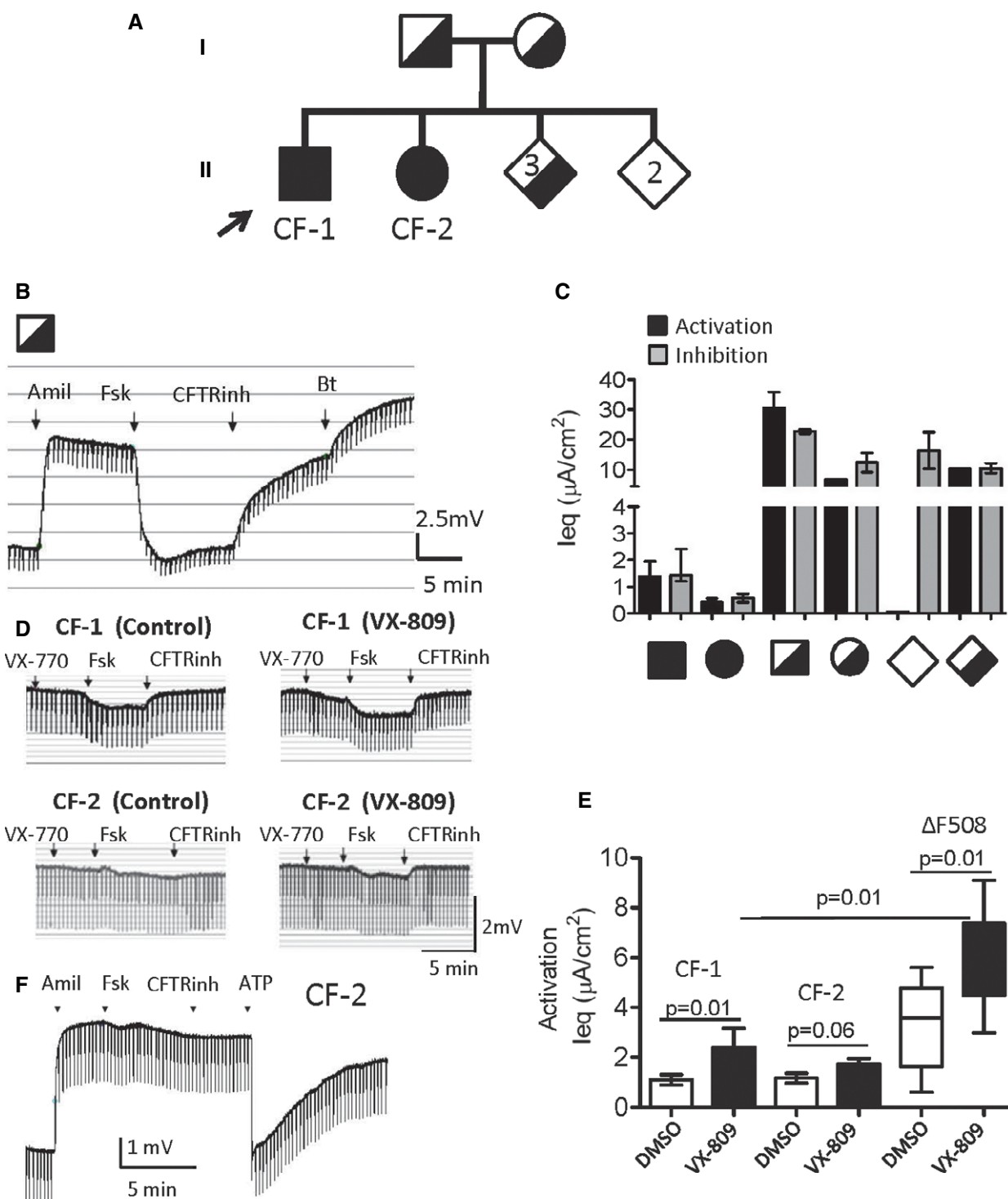

**Figure 6.**

patient (Fig 8C). Interestingly, the functional response of mutant CFTR measured in nasal cultures obtained from the subject CF-2 remained poor after VX-809 correction and VX-770 potentiation even following pre-treatment with the amplifier: PTI-CH (Fig EV7). Clearly, there are additional patient-specific factors that must be considered in therapy development for this individual.

## Discussion

A strategy for therapeutic development in the case of rare CF-causing mutations has been described using c.3700 A>G (or ΔI1234_R1239) as an example. *In silico* studies effectively predicted the defects in protein conformation and channel function revealed

◀

**Figure 6.  Nasal epithelial cultures derived from siblings homozygous for the c.3700 A>G mutation exhibit low CFTR channel function relative to non-CF family members and modest rescue by VX-809 and VX-770.**

A  Pedigree of the relationship of family members toward the index patients and their genotype status. Relative generation levels are marked as I and II; the proband is marked with an arrow; heterozygote carriers are represented by a half-filled square (father), circle (mother), or diamond (three siblings), while homozygous WT (two siblings) or c.3700 A>G individuals (CF-1 and CF-2) are unfilled or filled, respectively.

B  A representative tracing shows Ussing chamber measurements (symmetrical chloride concentrations) of CFTR function in nasal cell cultures from the non-CF father.

C  Summary of CFTR-dependent activation (black, forskolin alone for non-CF individuals, forskolin plus VX-770 for CF patients) and inhibition (gray, CFTRinh172) responses for several family members (mean ± SEM).

D  Representative tracings show Ussing chamber measurements of transepithelial potential difference traces of CFTR function in nasal cell cultures from affected patients (CF-1 and CF-2) in the absence or presence of the small molecule corrector VX-809.

E  Bar graph showing mean (± SEM) forskolin and VX-770 (1 μM) activated Ieq for nasal cultures from CF-1 and CF-2 after pre-treatment with VX-809 (48 h, 3 μM); replicate measurements of $n$ = 6 different cultures (CF-1) and $n$ = 5 (CF-2). The VX-809 and VX-770 rescued function for CF-1 remains significantly less than that measured in similarly treated nasal cultures derived from CF patients homozygous for ΔF508 (nasal cultures from six different patients donating to CFIT (CF Canada and SickKids Foundation sponsored program, box and whisker plots; middle lines indicate the median, box ranges indicate the lower and upper quartiles, and whiskers indicate the lowest and highest values)). Statistical significance of comparisons assessed using unpaired $t$-tests.

F  Representative tracing shows Ussing chamber measurements of calcium-activated chloride channel (CaCC) activity in nasal cell cultures from CF-2 in response to the P2Y2 receptor agonist ATP.

by studies of this mutant protein using overexpression in HEK-293 cells. Further, the HEK-293 expression system supported the potential efficacy of compounds currently approved for use in patients homozygous for the major mutation (lumacaftor and ivacaftor) in the functional rescue of this rare mutation. The combination of lumacaftor and ivacaftor (Orkambi®) did induce a significant increase in the functional expression of the rare mutant on the apical membrane of patient-derived nasal epithelial cultures; however, the response size was modest, falling short of a functional rescue observed in nasal cultures derived from individuals bearing the mutation for which Orkambi® was approved (i.e., ΔF508). The generation of a CRISPR/Cas9-edited bronchial epithelial cell line, engineered to express this mutant protein, enabled the identification of a companion therapy effective in improving the response to lumacaftor and ivacaftor. Importantly, these findings were functionally validated in iterative studies of patient-derived nasal tissues.

*In silico* and biochemical studies of the ΔI1234_R1239-CFTR revealed that it shares certain molecular defects with ΔF508-CFTR. The ΔI1234_R1239 mutation in NBD2 caused misassembly of the full-length protein, possibly by distortion of the NBD1:NBD2 interface, as well as allosterically reducing the conformational stability of the amino-terminal half of CFTR. Lumacaftor (VX-809) was

partially effective in correcting this assembly defect for the mutant protein overexpressed in HEK-293 cells, possibly via its proposed stabilizing effect on the interaction of ICL4 (conferred by MSD2) and NBD1 (Okiyoneda *et al*, 2013). Similarly, ivacaftor (VX-770) was partially effective in potentiating the channel activity of this mutant protein after its correction such that its activity was ~50% of that measured for "rescued" ΔF508-CFTR similarly expressed in HEK-293 cells. Together, these findings show that Orkambi® has the potential to partially repair the protein defects caused by this rare mutation. These preliminary studies prompted us to test the component drugs that make up Orkambi® in more relevant tissues bearing this rare CF-causing mutant.

To date, patient-derived nasal cultures have not been used regularly to assess the efficacy of corrector and potentiator compounds in the functional rescue of rare CF-causing mutations (Sabusap *et al*, 2016; Haggie *et al*, 2017). Hence, we were prompted to establish relevant benchmarks for normal CFTR function in nasal epithelial cultures. Specifically, we compared the CFTR channel activity in nasal epithelial cultures from CF family members to CFTR channel activity in nasal cultures derived from non-CF family members. Although it has been widely reported that there is considerable variability in *in vitro* CFTR-mediated responses among epithelial

**Figure 7.  HBE cells edited by CRISPR/Cas9 to express ΔI1234_R1239-CFTR recapitulate defects observed in patient-derived nasal epithelial cultures and provide a model for functional repair using novel small molecule amplifiers.**                                                ▶

A  Left panel: Immunoblot of steady-state expression of WT-CFTR in parental HBE cells and ΔI1234_R1239-CFTR (ΔIR) in gene-edited cells. CNX is used as a loading control. Right panel: Immunoblot of steady-state expression of ΔI1234_R1239-CFTR (ΔIR) in edited cells and patient-derived nasal epithelial tissue (CF-1). CNX is used as a loading control. Band B, black arrowhead; band C, white arrowhead.

B  Representative traces (membrane depolarization assay) of WT-CFTR, ΔI1234_R1239-CFTR (ΔIR), and CFTR knockout (KO) HBE cell lines; ΔI1234_R1239-CFTR function is measured following chronic treatment with VX-809. Black arrow, acute activation (forskolin + VX-770); gray arrow, acute inhibition (CFTRinh-172). Inset shows comparison of mean ± SEM change in membrane potential with activation in edited cells expressing WT-CFTR ($n$ = 4 platings, 4 technical replicates) or the mutant (ΔIR, $n$ = 9 platings, 4 technical replicates). Statistical significance of comparisons assessed by unpaired $t$-test.

C  ΔI1234_R1239-CFTR mRNA levels in HBE-ΔIR after 24-h treatment with vehicle (DMSO), PTI-CH (1 μM), or DMSO or PTI-CH in the presence of α-amanitin (50 μg/ml) to inhibit transcription (mean ± SEM, $n$ = 4). Statistical significance tested using two-way ANOVA with Tukey's multiple comparisons test.

D (i)  Immunoblot of steady-state CFTR expression levels in HBE-KO (DMSO) and HBE-ΔIR cells following 24-h treatment with vehicle (DMSO), VX-809 (3 μM), PTI-CH (1 μM), or VX-809 + PTI-CH. CNX was used as a loading control. Band B, black arrowhead; band C, white arrowhead.

D (ii)  Changes in band B or band C abundance after VX-809 plus PTI-CH pre-treatment were greater than those measured with VX-809 alone (mean ± SEM $n$ = 3 biological replicates, significance assessed using paired $t$-test).

E  Representative traces (membrane depolarization assay) of ΔI1234_R1239-CFTR function in HBE-ΔIR cells following 24-h treatment with vehicle (DMSO), VX-809, or VX-809 + PTI-CH. Black arrow, acute activation (forskolin + VX-770); gray arrow, acute inhibition (CFTRinh-172).

F  Quantitation (mean ± SEM) of peak responses to acute activation of ΔI1234_R1239-CFTR in HBE-ΔIR cells pretreated with DMSO ($n$ = 8), VX-809 alone ($n$ = 8) or VX-809 plus PTI-CH ($n$ = 4). Statistical significance of comparisons assessed using unpaired $t$-test.

cultures generated from CF patients, even across those harboring the same mutations (Van Goor *et al*, 2011; Awatade *et al*, 2015; Dekkers *et al*, 2015), we were surprised to observe the extent of large variability among the four non-CF family members. Despite this variability, the *in vitro* CFTR activity (CFTRinh-172-sensitive currents) in tissues from both CF individuals was much lower (~2–12%) than the lowest CFTR activity observed among the non-CF family members.

The *in vitro* studies of primary nasal cultures revealed that one of the two subjects (CF-2) in this family exhibited a lower level of total CFTR protein and residual CFTR channel function than the other (CF-1). Interestingly, the differential *in vitro* responses of these two subjects mirrored their relative respiratory health (Molinski *et al*, 2014). We confirmed that the poorer function exhibited in the cultures generated from patient CF-2 did not result from variable culture quality. Nasal cultures generated from both individuals expressed comparably levels of activity of another chloride channel, TMEM16A (Fig 5Bi). Further, we determined that there were no compound mutations in addition to c.3700 A>G in either subject. Both individuals (CF-1 and CF-2)

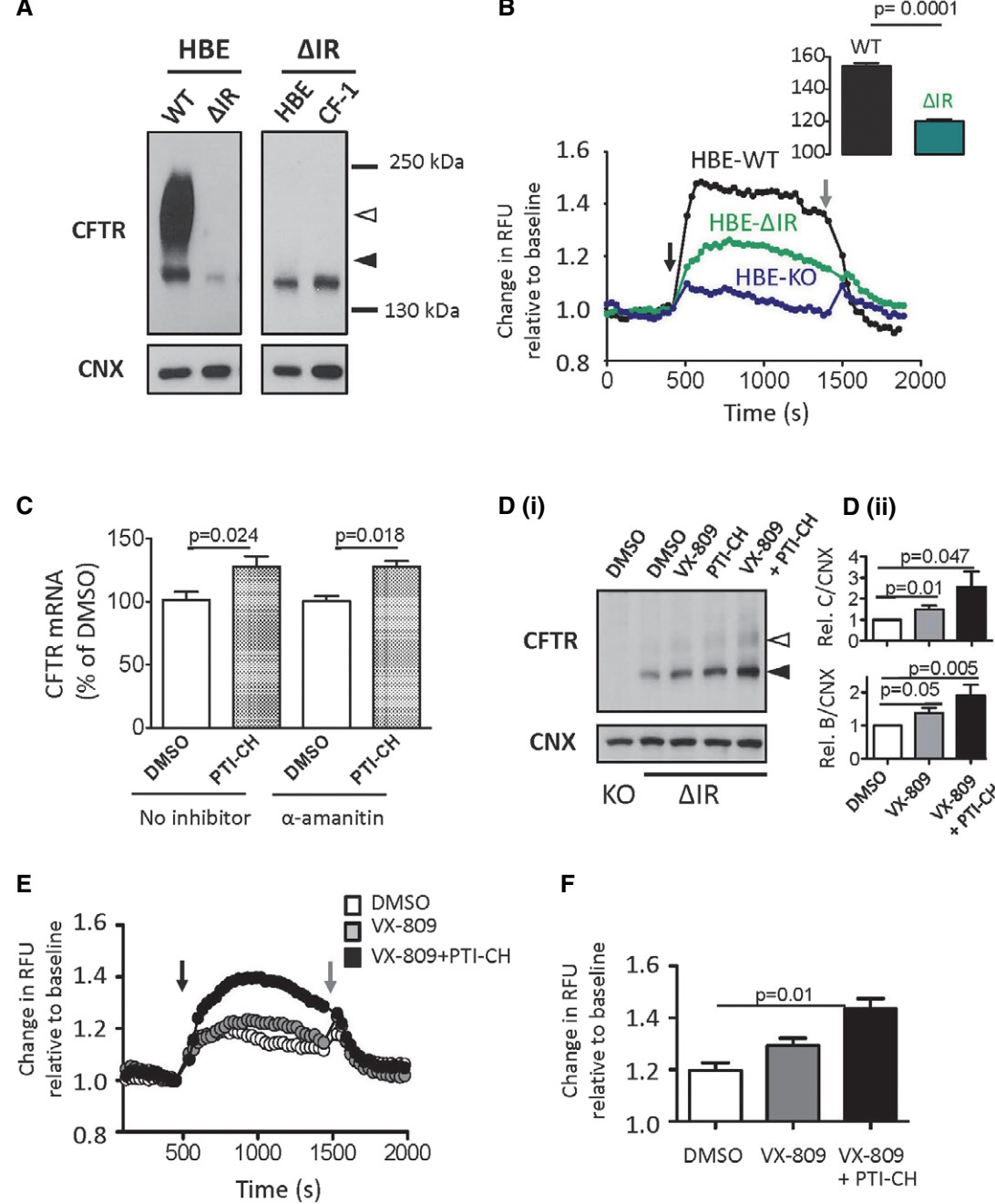

**Figure 7.**

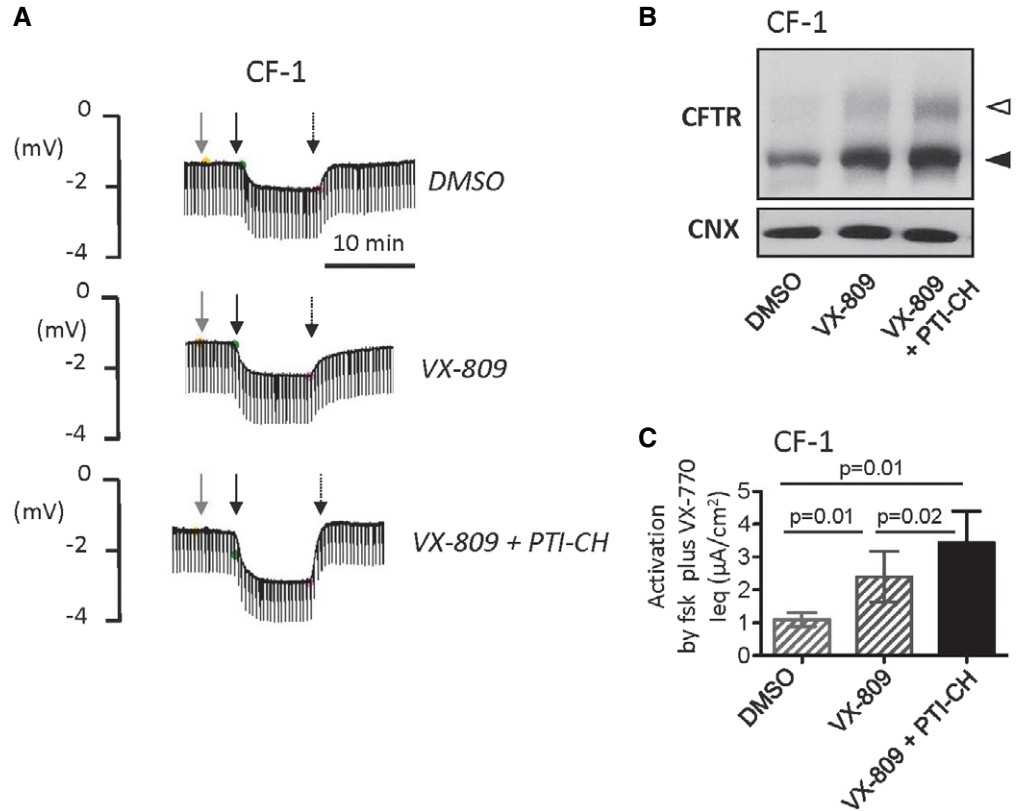

**Figure 8. Amplifying ΔI1234_R1239-CFTR expression enhances functional rescue effect of VX-809 + VX-770 in primary nasal cultures.**

A Ussing chamber responses (symmetrical chloride concentrations) exhibited by nasal epithelial cultures obtained from CF-1 after 24-h pre-treatment with vehicle (DMSO), VX-809, or VX-809 + PTI-CH. Responses to VX-770 (1 μM, gray arrow), forskolin (10 μM, thick black arrow), and CFTRinh-172 (5 μM, thin black arrow) were tested.

B ΔI1234_R1239-CFTR protein expression in nasal cultures from CF-1 after 24-h pre-treatment with vehicle (DMSO), VX-809 (3 μM), or VX-809 + PTI-CH (3 μM and 1 μM, respectively). Band B, black arrowhead; band C, white arrowhead; calnexin was used as a loading control.

C Summary of forskolin- and VX-770-dependent responses in Ussing chamber studies of nasal cultures from subject (CF-1). Bars represent mean and SEM of six biological replicates (different nasal cell seedings). The combination of VX-809 (3 μM) + PTI-CH (1 μM; black bar) significantly increased forskolin and VX-770 activated Ieq relative to VX-809 alone (hatched bars corresponding to primary nasal culture data shown previously in Fig 6E). Statistical significance of comparisons assessed using paired *t*-tests.

were tested by multiplex-PCR analysis for 39 recurrent *CFTR* mutations, by gene dosage using MLPA and by direct sequence analysis of the coding and flanking region of the *CFTR* gene [NCBI ref. NC_000007.13; NM_000492.3]. These tests confirmed that both individuals have identical *CFTR* mutation and polymorphism status (Table EV1). Hence, there are likely to be other genetic or environmental factors that determine the differential residual CFTR protein levels exhibited in nasal cultures from the two siblings bearing the c.3700 A>G mutation, factors yet to be determined.

Correction with VX-809 and potentiation with VX-770 increased the functional expression of the rare mutant CFTR, but this rescued function was modest at ~50% of the rescue measured in nasal cultures from *ΔF508* homozygous patients and < 20% of the minimum CFTR function observed in non-CF family members. Hence, these findings suggested that there is a need for a companion therapy to augment the effects of (Orkambi®) in patients with this rare mutation. We hypothesized, based on immunoblotting of primary nasal epithelial cultures, that one factor accounting for the poor

residual function and modest response to Orkambi® exhibited by nasal cultures from the affected individuals is the low abundance of mutant CFTR protein. We reasoned, on the basis of the studies shown in Fig EV3, that the low abundance of this mutant protein reflects in part, its degradation by the proteasomal pathway. We showed that the HBE cell line edited by CRISPR/Cas9 to express this mutant protein, faithfully modeled this defect in protein abundance, and proved instructive in informing future rescue strategies. Interventions aimed at increasing mutant protein expression, that is, the investigational compound that acts to amplify CFTR protein expression (PTI-CH), led to an improvement in the response to lumacaftor and ivacaftor, first in the edited cell line and finally—the more physiologically relevant patient-derived nasal culture model. Hence, our findings support the claim that this amplifier is in fact mutation-agnostic and may be considered for the treatment of certain CF-causing mutations that exhibit only partial rescue by correctors and potentiators.

This study highlights the importance of studying the consequences of CF-causing mutation in relevant tissues. While *in silico*

studies and biochemical experiments in heterologous expression systems provide insight into the potential for Orkambi® to rescue functional expression of this mutant CFTR protein, additional defects observed in the context of patient-derived tissues confer plausible secondary therapeutic targets. As shown here, an inherent deficiency in mutant protein abundance was revealed in patient-derived tissues, driving the development of a new CRISPR/Cas9-edited bronchial cell line that recapitulates this defect and will enable further therapy development.

# Materials and Methods

### Generation of ΔNBD2 mutant CFTR construct

E1172X-CFTR (ΔNBD2) was generated in human WT-CFTR cDNA (pcDNA3.1) using the KAPA HiFi HotStart PCR Kit (KAPA Biosystems, Woburn, MA) as previously described (Molinski *et al*, 2015). The NBD2 deletion construct was generated by introducing a stop codon at position E1172 using the following primers: 5′-GAC ATG CCA ACA TAA GGT AAA CCT ACC-3′ (sense); 5′-GGT AGG TTT ACC TTA TGT TGG CAT GTC-3′ (antisense). Plasmid DNA was prepared using the GenElute™ Plasmid Miniprep Kit (Sigma-Aldrich, St. Louis, MO), and the presence of the stop codon, as well as the integrity of CFTR cDNA, was confirmed by DNA sequencing (TCAG Inc., Toronto, ON).

### Studies of CFTR protein processing

The six-residue CFTR deletion mutant: ΔI1234_R1239-CFTR was generated in human WT-CFTR cDNA (pcDNA3.1) as previously described (Molinski *et al*, 2014). HEK-293 cells were transiently transfected with WT-CFTR or mutant CFTR constructs using PolyFect Transfection Reagent, according to the manufacturer's protocol (Qiagen, Hilden, Germany). HEK-293 cells transiently expressing CFTR proteins were maintained in DMEM (Wisent, St-Bruno, QC) supplemented with non-essential amino acids (Life Technologies, Waltham, MA) and 10% FBS (Wisent) at 37°C with 5% $CO_2$ (HEPA incubator, Thermo Electron Corporation; Molinski *et al*, 2014; Pasyk *et al*, 2015).

HEK-293 cells expressing WT-CFTR or mutant CFTR constructs were grown at 37°C for 24 h in the absence or presence of small molecules (3 μM VX-809 or 1 μM VX-661, Selleck Chemicals, Houston, TX; 10 μM C4, 6 μM C18, 10 μM VRT-325 or 10 μM VRT-532; Cystic Fibrosis Foundation Therapeutics) as required. Cells were subsequently lysed in modified radioimmunoprecipitation assay buffer (50 mM Tris–HCl, 150 mM NaCl, 1 mM EDTA, pH 7.4, 0.2% (v/v) SDS, and 0.1% (v/v) Triton X-100) containing a protease inhibitor cocktail (Roche) for 10 min, and the soluble fractions were analyzed by SDS–PAGE on 6% gels. After electrophoresis, proteins were transferred to nitrocellulose membranes and incubated in 5% (w/v) milk, and CFTR bands were detected using the human CFTR-NBD2-specific (Cui *et al*, 2007; amino acids 1,204–1,211) murine mAb 596 (1:20,000, University of North Carolina Chapel Hill, NC), using horseradish peroxidase-conjugated goat anti-mouse IgG secondary antibody (1:5,000), and by exposure to film for 0.5 to 5 min as required. Calnexin (CNX) was used as a loading control and detected using a CNX-specific rabbit Ab (1:5,000, Sigma-Aldrich), using horseradish peroxidase-conjugated goat anti-rabbit IgG secondary antibody (1:5,000), and by exposure to film for 0.5 to 5 min as required. Relative levels of CFTR proteins were quantitated by densitometry of immunoblots using ImageJ 1.42 Q software (National Institutes of Health), and reported values are normalized to CNX expression levels.

### Studies of ΔI1234_R1239-CFTR protein degradation

HEK-293 cells were transiently transfected with ΔI1234_R1239-CFTR at 37°C, as described previously (Molinski *et al*, 2014). Following transfection, HEK-293 cells overexpressing ΔI1234_R1239-CFTR were treated with 2 μM of the proteasomal inhibitor epoxomicin (Santa Cruz Biotechnology, Santa Cruz, CA), 100 nM of the lysosomal inhibitor bafilomycin A1 (Santa Cruz), or both, for 7 h at 37°C. Cells treated with either proteasomal, lysosomal, or both inhibitors were lysed following the respective treatments, and soluble proteins were analyzed by SDS–PAGE and immunoblotting as described previously (Molinski *et al*, 2014). The accumulation of ΔI1234_R1239-CFTR was compared in the presence of the treatments relative to vehicle (0.1% DMSO).

### Studies of CFTR-mediated fluorescence dequenching

To test function, HEK-293 cells overexpressing CFTR constructs were grown in 12-well plates, and upon formation of a mono-layer, the cells were incubated overnight with 10 mM of the halide-sensitive fluorophore 6-methoxy-N-(3-sulfopropyl)quinolinium (SPQ; Invitrogen Molecular Probes Inc.), at 37°C and 5% $CO_2$ as previously described (Molinski *et al*, 2014; Pasyk *et al*, 2015). Next day, the cells were washed three times with phosphate-buffered saline to make the extracellular fluid free of SPQ. Cells were then kept in chloride containing physiological solution, Hank's-buffered saline solution. Fluorescence measurements were made using Gemini EM Fluorescence microplate reader (Molecular devices). Intracellular SPQ was excited at 350 nm wavelength, and the emission was measured at 450 nm, reporting chloride concentration. After reading baseline fluorescence in presence of physiological solution, a chloride gradient was established via exchange of the extracellular solution with a chloride-free, nitrate-containing buffer ($NaNO_3$ 136 mM, $KNO_3$ 3 mM, $Ca(NO_3)_2$ 2 mM, glucose 11 mM, HEPES 20 mM, pH 7.2, and osmolarity 300 mOsm). CFTR was stimulated using cAMP agonist forskolin (10 μM). After reaching equilibrium for chloride flux, the extracellular solution was replaced with physiological solution containing chloride, allowing SPQ to quench. The fluorescence measurements were expressed as the change in fluorescence relative to the fluorescence measurement just before CFTR stimulation (Eckford *et al*, 2014; Molinski *et al*, 2014; Pasyk *et al*, 2015). Data are summarized as initial rate of change in relative fluorescence units in the first 5 min of stimulation. Each condition was repeated with at least two technical replicates on the same plate, and a total of at least three biological replicates per condition.

### Studies of CFTR-mediated membrane depolarization

CFTR-mediated membrane depolarization was measured as previously described (Maitra *et al*, 2013; Molinski *et al*, 2015). Briefly,

HEK-293 cells overexpressing CFTR constructs were grown to 100% confluence in 24-well plates (Costar) and washed with PBS, and the blue membrane potential dye (dissolved in chloride-free buffer containing 136 mM sodium gluconate, 3 mM potassium gluconate, 10 mM glucose, 20 mM HEPES, pH 7.35, 300 mOsm, at a concentration of 0.5 mg/ml; Molecular devices), which can detect changes in transmembrane potential, was added to the cells for 1 h at 27 or 37°C as required. The plate was then read in a fluorescence plate reader (Molecular devices—Gemini EM) at the required temperature, and after reading the baseline fluorescence, CFTR was stimulated using the cAMP agonist forskolin (10 μM; Sigma-Aldrich) or forskolin with VX-770 (1 μM; Selleck Chemicals). CFTR-mediated depolarization of the membrane was detected as an increase in fluorescence and repolarization or hyperpolarization as a decrease (Maitra *et al*, 2013). To terminate the assay, CFTR-specific inhibitor 172 (CFTRinh-172, 10 μM; Cystic Fibrosis Foundation Therapeutics) was added to all the wells. Changes in membrane potential were normalized to the addition of agonists.

## Preparation of crude membranes, and limited proteolysis of ΔI1234_R1239-CFTR

Crude membranes were prepared from HEK-293 cells transiently expressing WT-CFTR, ΔF508-CFTR, as well as ΔI1234_R1239-CFTR treated with either VX-809 (3 μM), Corr-4a (10 μM), VX-809 and Corr-4a (3 μM and 10 μM, respectively), or vehicle (0.1% DMSO) for 24 h at 37°C as previously described (Yu *et al*, 2011; Eckford *et al*, 2014). Briefly, cell pellets were resuspended in cell lysis buffer (10 mM HEPES, 1 mM EDTA, pH 7.2), and cells were lysed using a cell disruptor (10,000 psi, 4°C, 5 min). The cell suspension was centrifuged at 800 *g* for 10 min at 4°C to pellet unbroken cells, and crude membranes were isolated from the resulting supernatant after centrifugation at 100,000 *g* for 60 min at 4°C. The crude membrane pellet was resuspended in buffer (40 mM Tris–HCl, 5 mM MgCl₂, 0.1 mM EGTA, pH 7.4) by passage through a 1-ml syringe 20 times with a 27-gauge needle.

For limited proteolysis studies, 20 μg crude membranes were resuspended in buffer (40 mM Tris–HCl, 5 mM MgCl₂, 0.1 mM EGTA, pH 7.4) and sonicated. Samples were kept on ice, and trypsin (Promega) was added at the following concentrations: 0, 1.6, 3.1, 6.3, 12.5, 25, 50 μg/ml. Samples were again sonicated and incubated at 4°C for 15 min. Proteolysis was terminated by addition of 0.5 mg/ml trypsin soybean inhibitor (Sigma-Aldrich). Membranes were solubilized in modified RIPA for 15 min, and the soluble fraction was analyzed by SDS–PAGE on a 4–12% gradient gel. After electrophoresis, proteins were transferred to nitrocellulose membranes and incubated in Odyssey® blocking buffer (LI-COR Biosciences, Lincoln, NE), and protein bands were detected using the human CFTR-NBD1-specific murine mAb L12B4 (1:1,000, EMD Millipore, Billerica, MA) or the human CFTR-NBD2-specific murine mAb M3A7 (1:1,000, EMD Millipore). Fluorescence was detected using the secondary antibody IRDye-800 (goat anti-mouse IgG: 1:15,000, Rockland Immunochemicals, Gilbertsville, PA). Blots were imaged, and band intensities were detected using the Odyssey infrared imaging system (LI-COR Biosciences). Relative levels of full-length ΔI1234_R1239-CFTR resulting from trypsin digestion were measured using ImageJ 1.42 Q software (National Institutes of Health).

## Molecular dynamics system preparation and simulation conditions

The WT-CFTR homology model generated by Dalton and colleagues was embedded in a membrane-mimetic *n*-octane slab with an approximate thickness of 3.5 nm (Dalton *et al*, 2012). The system was $14.5 \times 12.4 \times 17$ nm³ in size with 2,511 octane molecules and was hydrated by 68,915 water molecules. A salt concentration of ~75 mM was achieved by adding 277 Cl⁻ and 258 Na⁺ ions to the aqueous solution. All calculations were performed using GROMACS version 4.5.5 (Hess *et al*, 2008) with the TIP3P water model (Jorgensen *et al*, 1983) and the OPLS-AA force field (Jorgensen *et al*, 1996) for protein and octane. The integration time step was 2 fs. A twin-range cutoff of 10 Å for van der Waals interactions and for direct electrostatic interactions calculated by particle-mesh Ewald (Essmann *et al*, 1995) was used, updating the neighbor list every 10 steps. Constant NPT conditions were applied using Parrinello-Rahman (Parrinello & Rahman, 1980) semi-isotropic pressure coupling in the plane of the membrane, with a constant pressure of one bar applied via a coupling constant of $\tau_P = 2.0$ and zero compressibility in the direction of the membrane normal. The aqueous solution, octane, and the protein were coupled separately to a temperature bath at 300 K with a coupling constant of $\tau_T = 0.1$ ps using the Nosé-Hoover algorithm (Nosé, 1984). The LINCS algorithm was used to constrain bond lengths (Hess *et al*, 1997). The molecular graphics in Fig 1 were produced using Visual Molecular Dynamics (VMD; Humphrey *et al*, 1996).

The system was first energy minimized using steepest descent followed by an initial pre-equilibration phase of 5 ns with position restraints on protein backbone and water oxygen atoms with a force constant of 1,000 kJ/mol/nm². This procedure allowed octane molecules to relax around the protein and at the water–octane interface until the octane density was stabilized. The second equilibration phase included position restraints on protein backbone atoms C, Cα, and N for another 20 ns while relaxing the octane and water molecules around the protein. After pre-equilibration, 18 independent replicas for WT were set up by randomizing the starting velocities to initiate production runs each of which consisted of an unrestrained simulation of 40 ns long. Following a procedure detailed by Kulleperuma and colleagues to perform cluster analysis, three representative structures that correspond to times frames at 14 ns (2 structures) and 10 ns (1 structure) and that also belong to the most populated cluster and obtained after at 14 ns (2 structures) and 10 ns (1 structure) of simulation were chosen to generate the ΔI1234_R1239 mutation (Kulleperuma *et al*, 2013).

The six-residue (I1234_R1239) loop was manually deleted followed by an energy minimization to close the gap between residue numbers 1,233 and 1,240. The above-mentioned pre-equilibration phases were performed on the ΔI1234_R1239 mutant system. After pre-equilibration, 18 independent replicas (six for each of the three starting conformations) of the ΔI1234_R1239 mutant were set up by randomizing the starting velocities to initiate production runs each of which consisted of a 30 ns unrestrained simulation.

All simulations were generated using GROMACS version 4.5.5 (Hess *et al*, 2008) with the OPLS-AA force field (Jorgensen *et al*, 1996) for protein and octane and the TIP3P model for water

(Jorgensen *et al*, 1983). The integration time step was 2 fs. A twin-range cutoff of 10 Å for the van der Waals interactions and 10 Å for direct electrostatic interactions calculated by particle-mesh Ewald (Essmann *et al*, 1995) was used, updating the neighbor list every 10 steps. Constant NPT conditions were applied using Parrinello-Rahman (Parrinello & Rahman, 1980) semi-isotropic pressure coupling in XY directions, with a constant pressure of one bar applied via a coupling constant of $\tau_P = 2.0$ and zero compressibility in the z direction. The aqueous solution, octane, and the protein were coupled separately to a temperature bath at 300 K with a coupling constant of $\tau_T = 0.1$ ps using the Nosé-Hoover algorithm (Nosé, 1984). The LINCS algorithm was used to constrain bond lengths (Hess *et al*, 1997). Graphics-related MD simulations are generated using VMD (Humphrey *et al*, 1996).

### Generation and characterization of primary nasal epithelial cells from a family with c.3700 A>G (ΔI1234_R1239-CFTR)

Nasal brushing was performed on family members (after obtaining informed consent) by a research nurse with procedural experience. A 3-mm-diameter sterile cytology brush (MP Corporation, Camarillo, CA) was used. The inferior turbinate was visualized and the brush inserted into the nares and rotated to brush the turbinate. The brush was then placed in warm basal epithelial growth media (BEGM, Lonza, Walkersville, MD). Cells were dissociated with gentle agitation and seeded on a collagen-coated flask (P0 or passage number = 0). Cultures were maintained at 37°C in BEGM with antibiotics and an atmosphere of 5% $CO_2$ in air. Cells were subsequently expanded into a larger flask (defined as P1) and passaged once 70–80% confluent. To generate air–liquid interface cultures, P2 cells were seeded on collagen-coated Transwell inserts (6.5 mm diameter, 0.4 μm pore size) at a density of $10^5$ cells per insert. Unused cells were frozen for later use. Cells were maintained in BEGM but once confluent, the media was changed to air–liquid interface (ALI) with basal differentiation media (PneumaCult, StemCell Tech., Vancouver, Canada). Basal media was changed every day for the first week and then every 2 days subsequent to that. The apical surface was washed weekly with PBS. By 3 weeks, cells had a ciliated phenotype.

### Ussing chamber studies of primary nasal epithelial cells

Nasal cell transwells were mounted in a circulating Ussing chamber (Physiological instruments Inc. San Diego, CA), continuously perfused with buffer (126 mM NaCl, 24 mM $NaHCO_3$, 2.13 mM $K_2HPO_4$, 0.38 mM $KH_2PO_4$, 1 mM $MgSO_4$, 1 mM $CaCl_2$, 10 mM glucose) with symmetrical chloride concentrations, and gassed with 5% $CO_2$ and 95% $O_2$ to maintain at pH 7.4. Transepithelial voltage and resistance, following brief 1 μA current pulses every 30 s, were recorded in open-circuit mode. Results are presented as calculated equivalent short-circuit current (Ieq). Following inhibition of epithelial $Na^+$ channel with amiloride, CFTR function was assessed as Forskolin-activated current (Ieq-Fsk) and as CFTRinh-172-sensitive current (Ieq-CFTRinh-172) which followed Forskolin activation. To test efficacy of corrector compounds, nasal cells were treated for 48 hrs with VX-809 (3 μM) or VX-661 (3 μM) prior to experiments, VX-770 as potentiator compound was applied acutely during the experiments.

### Studies of CFTR protein expression in primary nasal epithelial cells

Nasal cultures were grown at 37°C for 48 h in the absence or presence of small molecules (3 μM VX-809, 1 μM VX-661, or 4 mM 4-phenylbutyrate, 4-PBA) as required. Cells were then lysed in modified radioimmunoprecipitation assay buffer (50 mM Tris–HCl, 150 mM NaCl, 1 mM EDTA, pH 7.4, 0.2% (v/v) SDS, and 0.1% (v/v) Triton X-100) containing a protease inhibitor cocktail (Roche) for 10 min, and the soluble fractions were analyzed by SDS–PAGE on 6% gels as described above and previously (Molinski *et al*, 2014; Pasyk *et al*, 2015). After electrophoresis, proteins were transferred to nitrocellulose membranes and incubated in 5% (w/v) milk, and CFTR bands were detected using the human CFTR-MSD1-specific (amino acids 25–36) murine mAb MM13-4 (1:200, Abcam, Cambridge, UK), using horseradish peroxidase-conjugated goat anti-mouse IgG secondary antibody (1:2,500), and by exposure to film for 0.5 to 30 min as required. CNX was used as a loading control and detected using a CNX-specific rabbit Ab (1:5,000, Sigma-Aldrich), using horseradish peroxidase-conjugated goat anti-rabbit IgG secondary antibody (1:5,000), and by exposure to film for 0.5 to 5 min as required. Relative levels of CFTR proteins were quantitated by densitometry of immunoblots using ImageJ 1.42 Q software (National Institutes of Health), and reported values are normalized to CNX expression levels.

### Immunofluorescence detection of CFTR

ALI-cultured nasal epithelial cells on filter membranes were cut into 1- to 2-mm pieces and embedded in OCT compound (Sakura Finetek). Cryosections were cut at 6–8 μm. They were than fixed in cold 90% methanol/10% PBS solution at −20°C for 5 min. For *en face* imaging, epithelial cultures were fixed in a mixture of 2% paraformaldehyde, 0.01% glutaraldehyde, in 0.1 M phosphate buffer, pH 7.2 for 1 h. The cell layer was then scraped from the filter membrane using a razor blade. The released cell patches were neutralized in 0.15 M glycine, 80 mM $NH_4Cl$, 0.1 M phosphate buffer, pH 7.2 for 10 min, washed three times with 0.15 M glycine in phosphate buffer, pH 7.2 for 5 min, and permeabilized in 0.2% Triton X-100, in 0.15 M glycine, 0.5% BSA, 0.1 M phosphate buffer, pH 7.2 for 20 min, with solution changing every 4 min. To block non-specific staining, the samples were incubated in 4% BSA in PBS for 30 min. Incubation with primary and secondary antibodies was done in the same blocking solution. DAPI was used as a nuclear stain. Images were captured with the Nikon ECLIPSE Ti inverted microscope and NIS-Elements 3.10 software for fluorescence imaging.

### Studies of TMEM16A protein expression in primary nasal epithelial cells

Nasal epithelial cells from confluent transwells (6.5 mm diameter, 0.4 μm pore size) were each lysed in 150-μl modified radioimmunoprecipitation assay buffer (50 mM Tris–HCl, 150 mM NaCl, 1 mM EDTA, pH 7.4, 0.2% (v/v) SDS, and 0.1% (v/v) Triton X-100) containing a protease inhibitor cocktail (Roche) for 10 min, and the soluble fractions (20 μl) were analyzed by SDS–PAGE on 6% gels as described above and previously (Molinski

*et al*, 2014; Pasyk *et al*, 2015). After electrophoresis, proteins were transferred to nitrocellulose membranes and incubated in 5% (w/v) milk, and TMEM16A bands were detected using the human anti-TMEM16A rabbit mAb SP31 (1:100, Abcam), using horseradish peroxidase-conjugated goat anti-rabbit IgG secondary antibody (1:2,500), and by exposure to film for 0.5 to 5 min as required. CNX was used as a loading control and detected using a CNX-specific rabbit Ab (1:5,000, Sigma-Aldrich), using horseradish peroxidase-conjugated goat anti-rabbit IgG secondary antibody (1: 5,000) and by exposure to film for 0.5–5 min as required. Relative levels of CFTR proteins were quantitated by densitometry of immunoblots using ImageJ 1.42 Q software (National Institutes of Health), and reported values are normalized to CNX expression levels.

### CRISPR/Cas9-editing of a human bronchial epithelial cell line to endogenously express ΔI1234_R1239-CFTR

The human bronchial epithelial (HBE) cell line: 16HBE (a generous gift from Dr. D.C. Gruenert (University of California, San Francisco, CA) was used as a parental template for CRISPR/Cas9-mediated gene editing to introduce the ΔI1234_R1239 mutation on both alleles of the *CFTR* gene using standard procedures (Applied StemCell Inc, Milpitas, CA). Briefly, 16HBE cells was transfected with two different sgRNAs targeting exon 22 and intron 23 and a ssDNA donor oligo with the target sequence deleted (sequence: AGATGA CATCTGGCCCTCAGGGGGCCAAATGACTGTCAAAGATCTCACAGCA AAATACACAGAAGGTGGAAATGCCATATTAGAGAACATTTCCTTCT CAGTGAGATTTGAACACTGCTTGCTTTGTTAGACTGTGTTCAGTAA GTGAATCCCAGTAGCCTGAAGCAATGTGTTAGCAGAATCTATTTGT AACATTATTA). Two gRNAs (g11 and g13) were chosen based on the predicted off-target profile and the proximity to the target site: CFTRg11: AGTGTTCCAAATCTCACCCTCNGG and CFTRg13: CTGG CCAGGACTTATTGAGANGG. After transfection with gRNA constructs, Cas9 and donor plasmid, cells were selected transiently using puromycin. Post-selection, cells were then recovered and cultured in 96-well plates at a density of 1 cell/well. Single-cell clones were amplified and genotyped by PCR and sequencing.

### Measurement of PTI-CH-mediated stabilization of CFTR mRNA in CRISPR/Cas9-edited HBE cell line

HBE cells edited by CRISPR/Cas9 to express ΔI1234_R1239-CFTR were allowed to reach confluency in 10-cm dishes prior to treatment. Cells were incubated for 24 h with PTI-CH at 1 μM along with 50 μg/ml of the transcriptional inhibitor α-amanitin (Sigma-Aldrich). Total RNA was isolated using Trizol following the manufacturer's protocol (ThermoFisher, Waltham, MA). RNA concentrations were determined using a NanoDrop 2000/2000c Spectrophotometer (ThermoFisher). Total RNA (1 μg) was diluted into 26.4 μl of RNase-free water, and master mix was prepared using the following reagents: 4 μl 10× RT buffer, 4 μl 10× random primers, 1.6 μl 25× dNTPs, 2 μl RNasin (20 U/l), and 2 μl MultiScribe RT enzyme. Thermal cycler settings for the reverse transcription were as follows: 25°C for 10 min, 37°C for 120 min, and 85°C for 5 min. The qPCR (total volume of 20 μl reaction) was set up using 12 μl of Master Cocktail mix and 200 ng cDNA diluted in 8 μl RNase-free water. The Master Cocktail mix contained 10 μl Taqman fast advanced mix

**The paper explained**

**Problem**
Two drugs have been approved for the treatment of CF patients bearing specific *CFTR* gene mutations. Unfortunately, 50–60% of the total CF patient population does not yet have access to approved drugs because of the nature of their mutation. A major challenge in the field is to determine how many other patients, bearing rare mutations, could benefit from the approved drugs. *In silico* and biochemical studies of such mutant proteins after heterologous expression are effective in defining the molecular basis for disease and the potential response to mechanism-based therapies. Yet, it is well recognized in the field that the drug responses observed in heterologous expression systems are not always recapitulated in patient-derived tissues with fidelity. Hence, relevant cellular models of rare CF-causing mutations are required to enable effective testing of approved drugs (i.e., lumacaftor and ivacaftor, or Orkambi®) and to evaluate companion therapies with the potential to augment the efficacy of approved drugs.

**Results**
We performed *in silico* and biochemical studies of a rare CF-causing mutation: c.3700 A>G that results in an in-frame deletion of six residues in the second nucleotide binding domain of the CFTR protein (ΔI1234_R1239-CFTR). This mutant exhibited altered intramolecular interactions, defective assembly during biosynthesis, misprocessing and defective channel function. Certain defects were partially repaired using compounds (lumacaftor and ivacaftor) approved for the treatment of patients homozygous for the major mutant: ΔF508-CFTR, in a heterologous expression system. Yet, the rescue with this treatment was modest in nasal epithelial cultures derived from siblings homozygous for this mutation, failing to achieve 30% of the CFTR-mediated apical conduction measured in nasal cultures from non-CF family members. Interestingly, the nasal culture studies revealed an additional defect, namely reduced expression, a defect that was recapitulated in a bronchial epithelial cell line, edited to express ΔI1234_R1239-CFTR by CRISPR/Cas9. Importantly, we found that interventions aimed at enhancing CFTR protein abundance improved the rescue effect of lumacaftor and ivacaftor, first in the edited cell line and finally in patient-derived nasal epithelial cultures.

**Impact**
These findings strongly support the utility of CRISPR/Cas9-edited epithelial cell lines in testing the efficacy of companion therapies required to boost the response of rare CF-causing mutant proteins to CFTR modulators previously approved for the major mutant: ΔF508-CFTR.

(ThermoFisher) and 1 μl each of a CFTR probe set (ThermoFisher; catalogue # Hs00357011_m1) and actin as a reference probe set (ThermoFisher; catalogue # Hs01060665_g1). The reaction mix was dispensed into a qPCR compatible 384-well plate using a multichannel Matrix repeat pipette. All reactions were performed in duplicate to allow statistical analysis. The plate was sealed, vortexed briefly, and spun at 135 *g* for 1 min. The qPCR settings were as follows: 50°C for 2 min, 95°C for 20 s, and 40 cycles of 95°C for 1 s and 60°C for 20 s. When the qPCR program was complete, data were exported in an Excel format, analyzed using standard dCT and ddCT calculations, and graphed in GraphPad Prism.

### Statistical analysis

All data are represented as mean ± SEM unless otherwise noted. Prism 4.0 software (GraphPad Software, San Diego, CA) was used for statistical analysis. Paired *t*-test, unpaired Student's *t*-tests, one-way

analysis of variance (ANOVA), and two-way ANOVA were conducted as appropriate. The statistical test employed and *P*-values are indicated in each figure or figure legend. "N" values represent biological replicates (different samples or platings) and are indicated in the figure legends.

**Expanded View** for this article is available online.

## Acknowledgements

We thank the family of study participants for their contributions. The Research Ethics Board of SickKids Hospital (Toronto, Canada) approved the study, and informed consent was obtained to generate airway cultures from nasal brushings for the purpose of functional testing. The electrophysiological measurements obtained from nasal cultures generated from the individuals homozygous for ΔF508 were provided through the Program in Individualized Cystic Fibrosis Therapy (CFIT), supported by Cystic Fibrosis Canada and SickKids Hospital Foundation. The authors also thank Dr. Luis Galietta (U.O.C. Genetica Medica, Istituto Giannina Gaslini, Genova, Italy) for helpful comments regarding the TMEM16A studies, as well as Drs. Johanna Rommens and Roman Melnyk for helpful commentary on the manuscript. We thank Jennifer Curran for technical support. These studies were supported by Operating Grants to C.E.B. from the Canadian Institutes of Health Research (CIHR MOP-97954, CIHR GPG-102171) and Cystic Fibrosis Canada; New Investigator grant to T.G. from the Canadian Institutes of Health Research and Sick Kids Foundation; New Investigator grant to T.J.M. from the Canadian Institutes of Health Research and Sick Kids Foundation; Operating Grants to R.P. from the Canadian Institutes of Health Research (CIHR MOP-130461). S.V.M. was supported by Peterborough K.M. Hunter and Ontario Graduate Studentships. S.A. was supported by the H.W.C. Clayton Paediatric Research Studentship and Dr. Albert and Dorris Award for Cardiovascular Physiology. Computations were performed on the GPC super-computer at the SciNet HPC Consortium. SciNet is funded by the Canada Foundation for Innovation under the auspices of Compute Canada; the Government of Ontario; Ontario Research Fund—Research Excellence; and the University of Toronto. These studies were also partially funded by the Al Qamra Holding Group, located in Qatar, where c.3700 A>G affects most CF patients.

## Author contributions

The overall design of the study was by SVM and CEB; SVM, SA, WI, HO, AV, JPM, P-SL, KK, KD, MDP, PDWE, OL, LJH, LW, EL, PNR, RP, TJM, TG, FR, and CEB performed experiments, analyzed, and/or interpreted the results. The manuscript was primarily written by SVM and CEB with input from all authors.

## Conflict of interest

AV, JPM and P-SL were employed by Proteostasis Therapeutics during the course of the study and are shareholders of the company. Proteostasis Thera-peutics did not have any additional role in the study design, data collection and analysis, decision to publish, or preparation of the manuscript. All other authors declare that they have no competing interests.

## For more information

CFTR1: http://www.genet.sickkids.on.ca/cftr
CFTR2: http://www.cftr2.org

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
