## [Review Process File · EMBO Molecular Medicine]

ORKAMBI[®] and amplifier co-therapy improves function from a rare CFTR mutation in gene-edited cells and patient tissue

Steven V. Molinski, Saumel Ahmadi, Wan Ip, Hong Ouyang, Adriana Vilella, John P. Miller, Po-Shun Lee, Kethika Kulleperuma, Kai Du, Michelle Di Paola, Paul D.W. Eckford, Onofrio Laselva, Ling Jun Huan, Leigh Wellhauser, Ellen Li, Peter N. Ray, Régis Pomès, Theo J. Moraes, Tanja Gonska, Felix Ratjen and Christine E. Bear

Corresponding author: Christine Bear, Hospital for Sick Children

Review timeline:

Submission date:	03 October 2016
Editorial Decision:	11 October 2016
Authors' Appeal:	08 November 2016
Editorial Decision:	09 November 2016
Revision received:	05 February 2017
Editorial Decision:	24 March 2017
Revision received:	14 April 2017
Editorial Decision:	17 May 2017
Revision received:	23 May 2017
Accepted:	06 June 2017

Transaction Report:

Editor: Roberto Buccione

1st Editorial Decision

11 October 2016

Thank you for the submission of your manuscript "CRISPR/Cas9-edited cells and patient tissue inform rescue strategy for rare Cystic Fibrosis mutations" and apologies for the delay in replying due to the increased submission rate and because I sought advice from an external expert who was not immediately available.

I have now had the opportunity to read your paper and the related literature and I have also discussed it with my colleagues and the above-mentioned external expert. I am afraid that we concluded that the manuscript is not well suited for publication in EMBO Molecular Medicine and have therefore decided not to proceed with peer review.

Based on in vitro experimentation you first establish that the protein defect caused by I1234_R1239 CFTR mutation is potentially amenable to rescue by Orkambi, which however was not confirmed when tested on primary nasal cultures generated from patients with this mutation. You then find that a CRISPR/Cas9 edited bronchial epithelial cell line bearing this mutation was able to recapitulate the poor rescue effect observed in patient derived tissue. We appreciate that the CFTR amplifier PTI-CH appeared to improve the response to Orkambi up to a potentially acceptable therapeutic range.

We agreed with our advisor that the study is interesting but also concluded that the translational angle (very important for us and in general in this manuscript) is not conclusive and lacking important information. Indeed I found it exceedingly difficult, and ultimately failed to retrieve information on the nature of PTI-CH, its mechanism of action, and who and wherefrom it was obtained. I could actually find no mention of it in the literature (apart from the citation which only points to an abstract that is not readable anyway).

In conclusion, due also to the above concerns, we are not persuaded that your manuscript provides the striking level of conceptual advance and direct and novel clinical implications and/or translational development we would like to see in an EMBO Molecular Medicine article.

I do wish to add that, considered the potential interest of these findings, we would have no objection to consider a new manuscript on the same topic if at some time in the near future you have obtained data that would considerably strengthen the message of the study including through the provision of detailed analysis of PTI-CH's mode of action, pharmacology, characteristics etc.

I am sorry that I could not bring better news.

1st Revision - authors' response

05 February 2017

Thank you for the opportunity to resubmit our paper after additional experimentation and changes to the text. Now- we provide evidence for the mechanism of action of the amplifier (PTI-CH) on the rare mutation studied in this paper.

As you suggested, we conducted experimentation to:

- 1) provide evidence that PTI-CH's mode of action on the $\Delta I1234_R1239$ mutation is as reported in the Dukovski manuscript for the $\Delta F508$ mutation.
- 2) We re-wrote aspects of the manuscript (including the title) to highlight that this is the first successful use of a CFTR amplifier on a rare CF-causing mutation.
- 3) We also attached the submitted Dukovski manuscript for the reviewers to peruse and it is cited in our revised manuscript.

We look forward to hearing your thoughts and those of our reviewers regarding the revised manuscript

3rd Editorial Decision

24 March 2017

Thank you for the submission of your manuscript to EMBO Molecular Medicine. We have now heard back from the Reviewers whom we asked to evaluate your manuscript.

I again apologise for the significant delay in reaching a decision on your manuscript. In this case, we first experienced significant difficulties in securing Reviewers. Further to this the evaluations were delivered with delay.

I am afraid that in aggregate, the reviewer evaluations do not paint a positive picture. In fact, although reviewer 1 is rather positive (but very cursory), reviewers 2 and 3 raise fundamental and mostly overlapping concerns that question the suitability of this study for publication in EMBO Molecular Medicine.

Indeed, while the study is clearly considered potentially interesting, a core finding is questioned, namely whether the amplifier PTI-CH has any augmentative effect at all. This is a crucial point as the combination treatment with the amplifier was one of the main items of interest, which initially made me decide to send the manuscript out for peer-review. Reviewers 2 and 3 also criticise the poor quality of presentation, lack of crucial controls (e.g. $c3700A>G$ vs. $F508del$ comparison) the lack of essential information to ensure reproducibility (an issue close to our hearts at EMBO Press) and, in general, insufficient experimental support for the conclusions.

Although reviewer 2 suggests "toning down" some claims, this would not be an acceptable way forward for us.

After further editorial discussion, it was agreed that to address the concerns and to bring the manuscript to a sufficient level for publication in EMBO Molecular Medicine would require a major experimental undertaking, most likely not achievable within a 3-4 months timeframe and with no guarantee of the outcome. Considering that it is our policy to allow only one round of revision, I have no choice but to return the manuscript to you at this stage.

I wish to add that, considered the potential interest of these findings, we are open to considering a new manuscript on the same topic if at some time in the near future you have obtained additional data that would considerably strengthen the message of the study and address the Reviewers' concerns in full.

I am sorry to have to disappoint you at this stage, and hope the Reviewer comments are useful for your continued work on the topic.

***** Reviewer's comments *****

Referee #1 (Comments on Novelty/Model System):

This study provides important insights on a current question of high relevance to the therapeutic development of corrector molecules for the treatment of cystic fibrosis patients with class II mutations affecting CFTR trafficking and function. Although it was initially predicted that all class II CFTR mutations would respond to corrector molecules such as lumacaftor, the current study demonstrates that this is not necessarily the case. This paper provides a very detailed analysis of a rare CF causing mutation ($\Delta I1234_R1239$ -CFTR) impact on the CFTR protein and its response to Orkambi, the only combination of corrector/potentiator available to CF patients with the F508del mutation. They authors demonstrate a low correction by lumacaftor for this mutation as opposed to the canonical class II mutation F508del. They propose a new combination of drugs to efficiently restore $\Delta I1234_R1239$ -CFTR expression and function. Moreover, they authors demonstrate the interest and relevance of in vitro models of epithelial cells derived from patients.

Referee #1 (Remarks):

This is an excellent paper reporting extremely detailed and well thought study. It has many important points of relevance to CF and the study of the CFTR protein as well as therapeutic relevance. This reviewer very enthusiastically recommends publication.

Referee #2 (Comments on Novelty/Model System):

The main goal of this manuscript is to investigate the potential of Orkambi to correct the CFTR function of the c3700A>G mutation. In addition, the authors aimed to investigate the potential effect of a CFTR amplifier to enhance the response of this mutation to Orkambi. Their conclusion is that Orkambi by itself does not affect the CFTR c3700A>G function, however the amplifier augments the effect and reached the minimal level of WT CFTR.

The manuscript is potentially interesting, as it emphasize the need to classify CF patients carrying rare mutations according to their response to a treatment (Theratypes) and not by classes of mutations. This is highly important for other genetic diseases as well. Another potential interesting aspect of the work is the need to generate primary respiratory epithelial cells to study the effect of drugs, as the results in model systems overexpressing the mutations not always are recapitulated in cells from patients.

However, the manuscript in its current form has major drawbacks including:

1. The majority of the experiments are aiming to compare between the effect of drugs on the

c3700A>G and the F508del mutations since these two mutations were classified to the same class, misfolded CFTR protein, however the results for the F508del mutation are missing. The comparison relies on other studies described in the literature, however for such a comparison the experiments should have been performed in parallel with the same cellular systems and at the same conditions (for example in Figures 2B, 2C, 3A). This restricts the authors' ability to draw comparative conclusions. The authors should either perform the F508del experiments in parallel to the c3700A>G, or refer only to their results of the c3700A>G.

2. The effect of the PTI-CH, is very limited. The authors conclude that the PTI-CH augments the effect of VX-809 and VX-770 by upregulating CFTR mRNA level, the matured CFTR protein and its function. However, as can be seen in Figure 7F in the CRISPR edited HBE cells, there is a marginal effect on the mRNA as measured by qRT-PCR. This difference is below the resolution/accuracy of this technique. The effect on the matured CFTR protein level is also very weak (Figure 7C). The effect on the function, as measured by the depolarization assay (Figure 7D) lacks the comparison to the WT response.

Moreover, the analyses of the nasal epithelial cells from the patients show a limited functional effect: no significant effect is found in patient CF-2 and in patient CF-1 out of 6 repeated experiments, in 4 there was no significant effect and only in two there was an effect (Figure 8C). In Figure 8A, no CFTR band B is visible, only band C. How can this be explained? Altogether, the conclusion that the combination of Orkambi and the amplifier rescues the CFTR function are not substantiated by the results. Therefore, the conclusions should be modified to say that there is some effect in some of the experiments. This change is required in the text: Abstract, Results and Discussion as well as in the title of the manuscript.

3. The data presentation is sloppy, inconsistent, unclear and lacks many details. The experimental design is very difficult to follow. For example: is the assay in figures 3A, 4C and 7D the same? Is it the depolarization assay? If so, the different figures should be presented in the same format - same scale, including the effect of the CFTR inhibitor, etc. Another example: in Figure EV3E, how were the results calculated? relative to the DMSO. There are many more examples. In Figure EV3 F, G and H are missing. The result sections, the figures and figure legends should be rewritten, present clearly and consistently, with all the required experimental design details.

Referee #3 (Comments on Novelty/Model System):

The tools used to address the scientific question are state of the art, including in silico modeling, heterologous expression, expanded primary human airway cells and CRISPR/Cas9 gene editing. this comprehensive approach is novel. the ability to introduce mutations into a cell line to guide studies in primary brushed cells from patients is novel

Referee #3 (Remarks):

this is a very interesting and novel study to understand the mechanism of a rare CF disease causing mutation, and utilize state of the art tools to examine different restorative strategies.

Major comments:

1. the studies addressing impact of 4-PBA and PTI-CH on CFTR mRNA levels and function are not fully convincing that the mechanism of action is via mRNA stabilization. The mRNA increase produced by 4-PBA exceeds that of PTI-CH (figure EV6 vs Figure 7) but this doesn't translate into increased functional restoration. The non-specific nature of 4-PBA and the lack of data accompanying PTI-CH about mechanism of action makes rectification of these results difficult. furthermore, the change in CFTR mRNA levels is ~25% above untreated. The reviewer questions if this is sufficient change to impact protein levels to the observed effect.

2. ENaC is a critical part of CF pathology. Was there any impact of the modulator conditions from figure 8 on ENaC?

3. the data from Figure 8, panel C for CF-1 shows that 2:6 cultures differed in the VX809+PTI-CH conditions vs the controls, and there was no difference in CF-2. the reviewer questions whether this

is sufficient to draw a conclusion of drug effect. Furthermore the wide spread of data (non-CF controls, CF-1 vs CF-2) raises the question if the effects are due to drug effects, patient-specific effects or unique culture specific effects. Is there any data regarding the reproducibility of the culture techniques within subjects?

Minor comments:

1. The TMEM16A data is interesting, but could be moved to the supplement if space is an issue.
2. Page 21, para 2. This reviewer is not convinced with the argument regarding the relative importance of residual CFTR function to the variability in CF-1 vs CF-2 clinical disease manifestations. Additional data would be needed to support this conclusion (eg: sweat chloride, possibly NPD, pulmonary and GI clinical features). Furthermore, page 22, para 1 seemingly contradicts this hypothesis, discussing the role of potential genetic modifiers in contributing to the variable cell performance and patient phenotype. Finally, without repeatability data, it's hard to say whether this just represents sample to sample variability rather than true patient-specific differences. Additional data is needed to better understand what factors may be at play here.

2nd Revision - authors' response

14 April 2017

Responses to Reviewer #2

1. The majority of the experiments are aiming to compare between the effect of drugs on the c3700A>G and the F508del mutations since these two mutations were classified to the same class, misfolded CFTR protein, however the results for the F508del mutation are missing. The comparison relies on other studies described in the literature, however for such a comparison the experiments should have performed in parallel with the same cellular systems and at the same conditions (for example in Figures 2B, 2C, 3A). This restricts the authors' ability to draw comparative conclusions. The authors should either perform the F508del experiments in parallel to the c3700A>G, or refer only to their results of the c3700A>G.

We will include new Ussing chamber data from nasal cultures obtained from six individuals homozygous for F508del in our revised manuscript. The data clearly shows that nasal cultures from individuals with c3700A>G do not respond to ORKAMBI as well as nasal cultures from individuals homozygous for F508del (Figure 6). Similar comparisons were presented for HEK-293 over-expressing either mutant (Figure 2 and 3).

2. The effect of the PTI-CH, is very limited. The authors conclude that the PTI-CH augments the effect of VX-809 and VX-770 by upregulating CFTR mRNA level, the matured CFTR protein and its function. However, as can be seen in Figure 7F in the CRISPR edited HBE cells, there is a marginal effect on the mRNA as measured by qRT-PCR. This difference is below the resolution/accuracy of this technique. The effect on the matured CFTR protein level is also very weak (Figure 7C). The effect on the function, as measured by the depolarization assay (Figure 7D) lacks the comparison to the WT response.

We respectfully disagree with the comments regarding our data shown in Figure 7. The responses to PTI-CH on mRNA were small –but statistically significant. This effect of PTI-CH was statistically significant in enhancing the effect of VX-809 on mature protein abundance (now quantified in a new figure, Figure 7.D.ii) in this CRISPR-Cas9 edited cell line.

We showed the significant effect of the amplifier in augmenting function on VX-809 pretreated cells using a fluorescence based assay (we recently published in NPJ Genomic Medicine: <http://rdcu.be/rd5n>). The rescued response after PTI-CH and VX-809 relative to Wt is shown in the revised figure 7F (wt function shown as text and stippled line).

Moreover, the analyses of the nasal epithelial cells from the patients show a limited functional effect: no significant effect is found in patient CF-2 and in patient CF-1 out of 6 repeated experiments, in 4 there was no significant effect and only in two there was an effect (Figure 8C). In Figure 8A, no CFTR band B is visible, only band C. How can this be explained? Altogether, the

conclusion that the combination of Orkambi and the amplifier rescues the CFTR function are not substantiated by the results. Therefore, the conclusions should be modified to say that there is some effect in some of the experiments. This change is required in the text: Abstract, Results and Discussion as well as in the title of the manuscript.

Patient to patient variability in drug responses have been documented extensively in studies of CF lung transplant tissue (such variability was documented in the first studies of VX-809 by Van Goor et al, PNAS USA, 2011). Studies of drug responses in nasal cultures are just emerging in the literature but patient-to-patient variability in drug response in nasal tissues will not be surprising to most in the field.

We don't know the source for variability for drug responses in different nasal cultures from the same individual so we have been repeating these studies to increase the "n". Studies of patient derived nasal epithelial cultures are challenging. Biological replicates were obtained over two years by separate scrapings of the nasal cavity. However, we realized that we needed to focus on samples from c.3700 A>G patients obtained from May 2016 onwards as different culture conditions were introduced at this time. So, we will include data from scrapings from patient CF-1 after 2016 (patient CF-2 did not agree to submit to multiple scrapings). The positive effect of PTI-CH persists. There is a significant rescue effect of ORKAMBI- only in the presence of PTI-CH. We will include these new supportive data regarding efficacy of PTI-CH in nasal cultures from patient CF-1 in a revised Figure 8. We can also include a cleaner western blot showing the effect of PTI-CH on mutant protein processing nasal cultures obtained from patient CF-1 in a revised Figure 8.

3. The data presentation is sloppy, inconsistent, unclear and lacks many details. The experimental design is very difficult to follow. For example: is the assay in figures 3A, 4C and 7D the same? Is it the depolarization assay? If so, the different figures should be presented in the same format - same scale, including the effect of the CFTR inhibitor, etc. Another example: in Figure EV3E, how were the results calculated? relative to the DMSO. There are many more examples. In Figure EV3 F, G and H are missing. The result sections, the figures and figure legends should be rewritten, present clearly and consistently, with all the required experimental design details.

We were surprised and disconcerted by these comments –but will make every effort to improve clarity where possible. For example- we changed the format for figure 7 to improve clarity. The supplementary data was streamlined to improve clarity.

Referee #3

Major comments:

1. the studies addressing impact of 4-PBA and PTI-CH on CFTR mRNA levels and function are not fully convincing that the mechanism of action is via mRNA stabilization. The mRNA increase produced by 4-PBA exceeds that of PTI-CH (figure EV6 vs Figure 7) but this doesn't translate into increased functional restoration. The non-specific nature of 4-PBA and the lack of data accompanying PTI-CH about mechanism of action makes rectification of these results difficult. Furthermore, the change in CFTR mRNA levels is ~25% above untreated. The reviewer questions if this is sufficient change to impact protein levels to the observed effect.

PTI-CH is specific in modifying CFTR mRNA stability (as validated in the supporting paper by Duvoski and colleagues), so we focused on this compound rather than sodium butyrate. In our revision, we removed the data regarding sodium butyrate to improve the clarity of presentation. Although the increase in CFTR mRNA associated with PTI-CH treatment is small- it is significant and translates to a significant enhancement in the VX-809 mediated rescue of this mutant protein. In this revision we include new quantification of mature CFTR protein Figure 7.D.ii

2. ENaC is a critical part of CF pathology. Was there any impact of the modulator conditions from figure 8 on ENaC?

This is an interesting question but broad and will need to be addressed in subsequent studies.

3. the data from Figure 8, panel C for CF-1 shows that 2:6 cultures differed in the VX809+PTI-CH

conditions vs the controls, and there was no difference in CF-2. the reviewer questions whether this is sufficient to draw a conclusion of drug effect. Furthermore the wide spread of data (non-CF controls, CF-1 vs CF-2) raises the question if the effects are due to drug effects, patient-specific effects or unique culture specific effects. Is there any data regarding the reproducibility of the culture techniques within subjects?

Since the submission of our paper, we conducted additional studies with nasal cultures from CF-1 (CF-2 has refused to allow additional scraping). The effect of PTI-CH persists and there is a significant rescue effect of ORKAMBI- only in the presence of PTI-CH. We will include these new supportive data regarding efficacy of PTI-CH in nasal cultures from patient CF-1 in figure 6 and figure 8. Please remember that there are only 16 of these patients listed in the CFTR2 database- it is an excellent example of a rare CF causing mutation and the need for personalized approaches.

Minor comments:

1. The TMEM16A data is interesting, but could be moved to the supplement if space is an issue.

We think that the TMEM16A data is important in validating the quality of the nasal cultures and should remain in the main body of the paper.

2. Page 21, para 2. This reviewer is not convinced with the argument regarding the relative importance of residual CFTR function to the variability in CF-1 vs CF-2 clinical disease manifestations. Additional data would be needed to support this conclusion (eg: sweat chloride, possibly NPD, pulmonary and GI clinical features). Furthermore, page 22, para 1 seemingly contradicts this hypothesis, discussing the role of potential genetic modifiers in contributing to the variable cell performance and patient phenotype. Finally, without repeatability data, it's hard to say whether this just represents sample to sample variability rather than true patient-specific differences. Additional data is needed to better understand what factors may be at play here.

We are happy to remove our speculation regarding CF-2 from our Discussion.

4th Editorial Decision

17 May 2017

Thank you for the submission of your revised manuscript to EMBO Molecular Medicine. We are sorry that it has taken longer than we would have liked to get back to you on your manuscript.

In fact, we were unable to obtain an evaluation from reviewer 3 and I had to therefore ask reviewer 2 to evaluate your revision also on his/her behalf. The inevitable consequent delay was also compounded with the fact that I wished to discuss this case further with my colleagues while also traveling.

You will see that although the reviewer is now globally positive, s/he lists a number of items for your action, including the toning down of some conclusions, reinserting data that had been omitted, and a few other items. I am prepared to make an editorial decision on the next, final version of your manuscript, provided you carefully and fully address the remaining concerns. Please highlight the changes in the manuscript.

To ensure prompt processing of your manuscript, please also comply with the following editorial requirements (<http://embomolmed.embopress.org/authorguide>):

- 1) Please add author information for all contributing authors
- 2) Please move the EV legends to the main manuscript file and upload individual figure files for the EV figures
- 3) We note that manuscript callouts for for Fig 5Bii and 5Biv are missing
- 4) Supplemental Table 1 should be renamed Table EV1 and the relative callout updated. Also, a DOC or XLS file should be provided for this table

5) As per our Author Guidelines, the description of all reported data that includes statistical testing must state the name of the statistical test used to generate error bars and P values, the number (n) of independent experiments underlying each data point (not replicate measures of one sample), and the actual P value for each test (not merely 'significant' or ' $P < 0.05$ ').

6) For experiments involving human subjects the authors must identify the committee approving the experiments and include a statement that informed consent was obtained from all subjects and that the experiments conformed to the principles set out in the WMA Declaration of Helsinki [<http://www.wma.net/en/30publications/10policies/b3/>] and the NIH Belmont Report [<http://ohsr.od.nih.gov/guidelines/belmont.html>]. Any restrictions on the availability or on the use of human data or samples should be clearly specified in the manuscript. Any restrictions that may detract from the overall impact of a study or undermine its reproducibility will be taken into account in the editorial decision.

I look forward to seeing a revised form of your manuscript as soon as possible.

***** Reviewer's comments *****

Referee #2 (Remarks):

I read the authors' response to the critics raised by me and by reviewer 3, as requested. My response refers to both.

Enhancement of the effect of CFTR drugs is highly important. The main aim of the study was to show the effect of PTI-CH in correcting the CFTR level and function in combination with correctors and potentiators, in cells carrying a rare mutation, c.3700A>G. In this respect, the results of the Western blots and the functional assays in the cell lines are convincing (Figure 7) and the success of establishing the CRIPR/CAS9- edited HBE cells is very important. However, the transcript changes in these cells are not convincing. A change of ~25% of a cycle is within the variability of duplicates and is below the resolution of the technique. Hence, these results should not be included in the manuscript.

Regarding the results in cells from the patients' cells. It is pity that the results of CF2 are not included in Fig 8 of the revised version, as variability in the response among patients is highly important. The results actually indicate that there is a real difference between patients C1 and C2 and that the difference is not only technical, as the authors raised, but rather reflect inherent cellular differences between the patients. In any case, the results of C2 are presented in figures 5 and 6 and were removed only from figure 7. It is important that the results in Figure 7 will include C2 as well and that the difference will be clearly discussed.

Figure 6 - How patient 2 in the revised version is no NS while its response was significant in the previous version?

In conclusion, the effect of PTI-CH on the transcript should not be included in the manuscript. In addition, although the effect of PTI-CH is promising, the tone of the conclusions is way too strong. The results do not support rescue of the function but rather improvement or enhancement. Hence, the conclusions in the Results, Discussion, Abstract and Title should be accordingly modified.

Minor comments

- Page 16 line 3 the ref to the figure is incorrect - should be 7D
- The scale in figure 7E is still different from the scale in all other figures presenting data based on this assay.

Response to Reviewer #2. We addressed all of the recommendations by Reviewer #2. In doing so we changed Figure 7 and edited the text. The revised text is highlighted yellow for ease in tracking changes.

Enhancement of the effect of CFTR drugs is highly important. The main aim of the study was to show the effect of PTI-CH in correcting the CFTR level and function in combination with correctors and potentiators, in cells carrying a rare mutation, c.3700A>G. In this respect, the results of the Western blots and the functional assays in the cell lines are convincing (Figure 7) and the success of establishing the CRIPR/CAS9- edited HBE cells is very important. However, the transcript changes in these cells are not convincing. A change of ~25% of a cycle is within the variability of duplicates and is below the resolution of the technique. Hence, these results should not be included in the manuscript.

Author Response: We downplayed the interpretation of the RNA results in the title, abstract and body of the text. All of these changes are highlighted in the revised manuscript.

Regarding the results in cells from the patients' cells. It is pity that the results of CF2 are not included in Fig 8 of the revised version, as variability in the response among patients is highly important. The results actually indicate that there is a real difference between patients C1 and C2 and that the difference is not only technical, as the authors raised, but rather reflect inherent cellular differences between the patients. In any case, the results of C2 are presented in figures 5 and 6 and were removed only from figure 7. It is important that the results in Figure 7 will include C2 as well and that the difference will be clearly discussed.

Author Response: The results of C2 (CF2) have been re-added as EV7. The Discussion has been revised to re-add our previous discussion regarding the possible causes for the variation between CF-1 and CF-2 and this test has been highlighted.

Figure 6 - How patient 2 in the revised version is no NS while its response was significant in the previous version?

Author Response: The response for the cultures from patients CF-2 were always marginal and with increased "n" this trend was confirmed as insignificant.

Although the effect of PTI-CH is promising, the tone of the conclusions is way too strong. The results do not support rescue of the function but rather improvement or enhancement. Hence, the conclusions in the Results, Discussion, Abstract and Title should be accordingly modified.

Author Response: The wording has been revised through the manuscript (including the title) as suggested by the reviewer.

Minor comments

- Page 16 line 3 the ref to the figure is incorrect - should be 7D
- The scale in figure 7E is still different from the scale in all other figures presenting data based on this assay.

Authors Response: These suggested changes were implemented in the revised manuscript

Corresponding Author Name: Christine E. Bear

Manuscript Number: EMM-2016-07137-V4